# A cholinergic-sympathetic pathway primes immunity in hypertension and mediates brain-to-spleen communication

Daniela Carnevale[1,2], Marialuisa Perrotta[1], Fabio Pallante[1], Valentina Fardella[1], Roberta Iacobucci[1], Stefania Fardella[1], Lorenzo Carnevale[1], Raimondo Carnevale[1], Massimiliano De Lucia[1], Giuseppe Cifelli[1] & Giuseppe Lembo[1,2]

The crucial role of the immune system in hypertension is now widely recognized. We previously reported that hypertensive challenges couple the nervous drive with immune system activation, but the physiological and molecular mechanisms of this connection are unknown. Here, we show that hypertensive challenges activate splenic sympathetic nerve discharge to prime immune response. More specifically, a vagus-splenic nerve drive, mediated by nicotinic cholinergic receptors, links the brain and spleen. The sympathetic discharge induced by hypertensive stimuli was absent in both coeliac vagotomized mice and in mice lacking α7nAChR, a receptor typically expressed by peripheral ganglionic neurons. This cholinergic-sympathetic pathway is necessary for T cell activation and egression on hypertensive challenges. In addition, we show that selectively thermoablating the splenic nerve prevents T cell egression and protects against hypertension. This novel experimental procedure for selective splenic denervation suggests new clinical strategies for resistant hypertension.

[1] Department of Angiocardioneurology and Translational Medicine, IRCCS Neuromed, 86077 Pozzilli, Italy. [2] Department of Molecular Medicine, 'Sapienza' University of Rome, 00161 Rome, Italy. Correspondence and requests for materials should be addressed to D.C. (email: daniela.carnevale@neuromed.it) or to G.L. (email: lembo@neuromed.it).

Despite the wide spectrum of antihypertensive medications currently available, a significant proportion of patients still have poorly controlled blood pressure and, consequently, resistant hypertension[1–3]. In the first half of the past century, the Mosaic Theory of Hypertension proposed that many factors, including genetics and environment as well as nervous, mechanical and hormonal perturbations, interplay to raise blood pressure[4]. Because the aetiology of hypertension is multifaceted, its solution must involve more than a single anatomical organ.

In the last decade, immunity emerged as a crucial player in hypertension: immune cells infiltrate the vessel walls and kidneys of hypertensive animals, and mice without lymphocytes are protected from angiotensinII (AngII)-induced hypertension[5–9]. Despite growing awareness of immune cells' role in hypertension, blood pressure control has been considered primarily the work of the autonomic nervous system[10]. The sympathetic nervous system, by contrast, has historically been thought to influence blood pressure control via regulation of key physiological parameters including heart rate, vascular tone and renal sodium excretion[3]. Clinically, sympathetic innervation of the kidney has been seen as the cause of hypertension, and renal denervation has been regarded as an intriguing new approach to treating uncontrolled high blood pressure[11–13].

The autonomic nervous system also regulates of immunity[14–17], though its role is often overlooked. Previously, we have found that placental growth factor (PlGF), an angiogenic growth factor belonging to the vascular endothelial growth factor (VEGF) family, links the nervous drive to immune system activation[7]. We performed a coeliac ganglionectomy to selectively disrupt sympathetic innervation of splanchnic circulation and found that PlGF activation, immune system recruitment and increased blood pressure depend on this pathway[7]. Interestingly, preliminary studies indicated that splanchnic innervation would impact hypertension[18]. However, two main questions remain unanswered: (i) which splanchnic nervous compartment is responsible for these effects on immunity and blood pressure regulation and (ii) how is the brain-to-splanchnic compartment connection established at the onset of hypertension.

## Results

**Hypertensive stimuli activate splenic nerve discharge.** Although chronic AngII has been shown to be a potent driver of splanchnic circulation[18], we do not yet know which downstream nervous district is engaged by hypertensive stimuli to increase blood pressure. The chronic effects of AngII are mediated by brain signals transmitted via the peripheral networks[19]. Indeed, lesions in central nervous system organs, which are able to sense peripheral challenges, prevent AngII infusion from raising blood pressure[19]. Taken together, these connections suggest that peripherally-delivered hypertensive stimuli activate a nervous drive in the brain that is conveyed through splanchnic circulation. Given that immune system activation in hypertension depends on the same pathway, we hypothesized that the splenic nerve could be the splanchnic innervation connecting the brain and immune cells.

To test this idea, we established a procedure for recording sympathetic nerve activity (SNA) in mice. First, we validated splenic nerve SNA (SSNA) recording by inducing reflex responses to blood pressure changes. We pharmacologically manipulated blood pressure with sodium nitroprusside and phenylephrine to

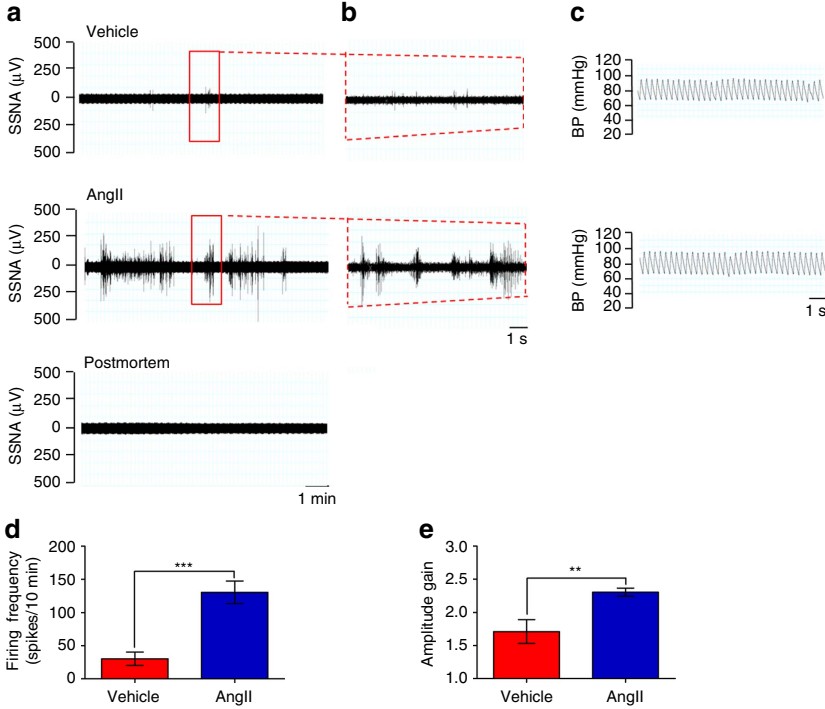

**Figure 1 | AngII activates discharge of the splenic nerve. (a,b)** Representative raw signals of SSNA in a time bin of 10 min in (**a**) show a significantly increased nerve discharge in AngII-infused mice as compared with vehicle. Raw signal of residual SSNA during postmortem was used to identify the signal threshold level for each recording. In (**b**) it is represented an enlarged view of 10 s of SSNA raw signal from the red window in **a**. (**c**) Blood pressure (BP) monitoring during SSNA recording excluded hemodynamic effects. (**d**) Firing frequency, represented by the mean number of spikes in a time bin, was significantly higher in AngII-infused mice as compared with vehicle (vehicle $n_{mice} = 8$ and AngII $n_{mice} = 10$; independent samples Student's $t$-test, $t(16) = -4.697$, \*\*\*$P < 0.001$). (**e**) Analysis of mean amplitude gain of spikes showed that AngII-infused mice had also increased SSNA in the pattern of burst amplitude as compared with vehicle (vehicle $n_{mice} = 8$ and AngII $n_{mice} = 10$; independent samples Student's $t$-test, $t(9) = -3.120$, \*\*$P < 0.01$).

evaluate typical SNA reflex responses (Supplementary Figs 1 and 2). We found that SSNA increased quickly in response to lowered blood pressure, as expected for sympathetic activity (Supplementary Fig. 1a–c). To further assess nerve activity, we evaluated synchronous sympathetic bursts. Analysis of SSNA showed that most nerve bursts synchronized with raw blood pressure signal (Supplementary Fig. 1d,e), a result that is consistent with the semirhythmic SNA observed in nerves in nearby compartments, including the splanchnic[10,20–22]. Ganglionic blockade with hexamethonium abolished SSNA activity (Supplementary Fig. 1f), which overlapped the residual postmortem recording (Supplementary Fig. 1g) that was used thereafter as the positive action potential threshold. As expected, the opposite response was observed when mice were challenged with phenylephrine to acutely increase blood pressure. Indeed, the SSNA, although similarly pulse synchronous, was markedly reduced (Supplementary Fig. 2a–g).

After validating the direct SSNA recording method, we tested whether chronic AngII infusion activates SSNA to modulate subsequent T cell activation. To avoid the potentially confounding effects of increased blood pressure itself, we analyzed mice 3 days after starting AngII infusion, when blood pressure has not yet significantly changed (Supplementary Fig. 3). To find the appropriate time point for examining the connection between nervous drive and immune activation, we performed a time course analysis of T cell recruitment in target organs. After 3 days, we could detect T cells in the aortas (Supplementary Fig. 4a,b) and kidneys (Supplementary Fig. 4c,d) of AngII-infused mice.

Next, we used a bipolar electrode to record SSNA in splenic nerve bundles in mice infused with either AngII or Vehicle (Fig. 1). A representative recording of splenic nerve discharge clearly shows increased rhythmic SSNA bursts above the background noise, as defined by the postmortem recording (Fig. 1a,b). Simultaneous blood pressure recording demonstrated animal stability during the procedure (Fig. 1c). Changes in SSNA included increases in either the number of sympathetic bursts or the amplitude of individual bursts. We found that AngII-infused mice had significantly higher SSNA, measured both as firing frequency (Fig. 1d) and as mean amplitude gain (Fig. 1e), than controls.

To determine if the increased SSNA was a selective response to AngII, we also performed experiments with deoxycorticosterone acetate (DOCA)-salt hypertensive mice, a model characterized by increased AngII levels in the brain and reduced peripheral renin–angiotensin system (RAS) activation[23]. As in the experiments detailed above, in this model we studied mice at the beginning of blood pressure increase (Supplementary Fig. 5) and again found elevated SSNA (Fig. 2a–c). Quantitative analysis confirmed that SSNA activation is similar in DOCA-salt-treated mice and AngII-hypertensive mice (Fig. 2d,e), thereby suggesting the brain-to-splenic nerve drive is a common pathway for hypertensive challenges that are sensed by the brain and consequently lead to elevated blood pressure.

**The vagal-coeliac-splenic connection mediates hypertension**. Sympathetic innervation of the spleen has been recognized as a crucial mechanism for modulating immune cell functions[24–26]. For many years, autonomic innervation of immune organs did not seem relevant to cardiology, but current evidence for immunity's vital role in blood pressure regulation necessitates investigating neuro-immune reflexes in hypertension. Sympathetic and parasympathetic innervations of the splanchnic organs have been traditionally regarded as anatomically separate along their peripheral routes. If the innervations are indeed separate, the splenic nerve should be activated by a pre-ganglionic neuron in the paravertebral ganglia chain that connects to the intermediolateral grey column in the spinal cord. However, immunologists have identified a cholinergic anti-inflammatory reflex that is mediated by the vagus nerve's coeliac branch and specifically targets splenic innervation during septic challenges[24–27].

To investigate whether and how vagal signals could affect splenic nerve activity during hypertensive challenges, we performed cervical vagotomy while recording SSNA (Fig. 3a–c). Cervical vagotomy completely prevented AngII from activating the splenic nerve (Fig. 3a,b). These results indicate that AngII-induced splenic nerve activation is mediated by the cervical vagus nerve. Unexpectedly, we found no reduction in residual SSNA gain amplitude after both cervical and coeliac vagotomy (Fig. 3c), a result that suggests hypertensive stimuli use different sympathetic pathways for SSNA firing frequency and individual burst amplitude. As previously reported concerning the regulation of other organs[10], our results indicate that a specific brain cell network produces rhythmical discharges in the splenic nerve through the vagus connection. Conversely, other central nervous system cell groups regulate splenic nerve gain amplitude independently from the vagus nerve. To further uncover the relative contributions of the vagus nerve's afferent versus efferent arms, we performed a coeliac vagotomy that also completely inhibited AngII-induced SSNA (Fig. 3d,e), still with no effect on the gain amplitude of splenic nerve activity (Fig. 3f). In addition, we analyzed the chronic response to AngII in mice with left coeliac vagotomy. Inhibiting vagus nerve efferents hampered the blood pressure increase typically induced by AngII (Fig. 4a,b) as well as the T cell egression (Fig. 4c,d).

Overall, these results support our proposal that, although the vagus nerve has been considered paradigmatic for parasympathetic modulation of blood pressure[28], it also activates the splenic nerve under hypertensive challenges. This previously unknown aspect of hypertension may indicate that completely different pathophysiological conditions, such as sepsis and hypertension, can affect the same neuro-immune connection.

**α7nAChR at the intersection of the vagus and splenic nerves**. Mammals regulate synaptic transmission across sympathetic and parasympathetic ganglia via nicotinic acetylcholine receptors (nAChR), which can assemble in various stoichiometries and therefore have many functions[29]. One nAChR, the α7 subunit, can form homomeric pentamers that mediate excitatory postsynaptic currents. Interestingly, α7nAChR also participates in the cholinergic anti-inflammatory reflex and, though initially discovered as a modulator of macrophage function, has been shown to couple the vagus-splenic connection at the neuronal level[25,26]. Yet α7nAChR is not required for the parasympathetic modulations exerted by the vagus nerve in typical cardiovascular parameters such as heart rate[30]. To our surprise, α7nAChR KO mice infused with AngII had significantly inhibited SSNA firing frequency (Fig. 5a–d), which suggests the drive was hampered before reaching the splenic nerve. As observed in vagotomized mice, AngII infusion did not alter SSNA amplitude gain in α7nAChR KO mice (Fig. 5e). It may be that different SNA patterns can evoke different neurotransmitter responses[10]. Because the AngII-infused vagotomized and α7nAChR KO mice showed the vagus-splenic connection affects firing frequency but not amplitude gain, we asked which pattern is relevant for noradrenaline release in the spleen. Since AngII increases noradrenaline in the spleen through the activation of tyrosine hydroxylase[7], we examined this response in AngII-infused α7nAChR KO mice and found that both noradrenaline content (Supplementary Fig. 6a) and tyrosine hydroxylase staining

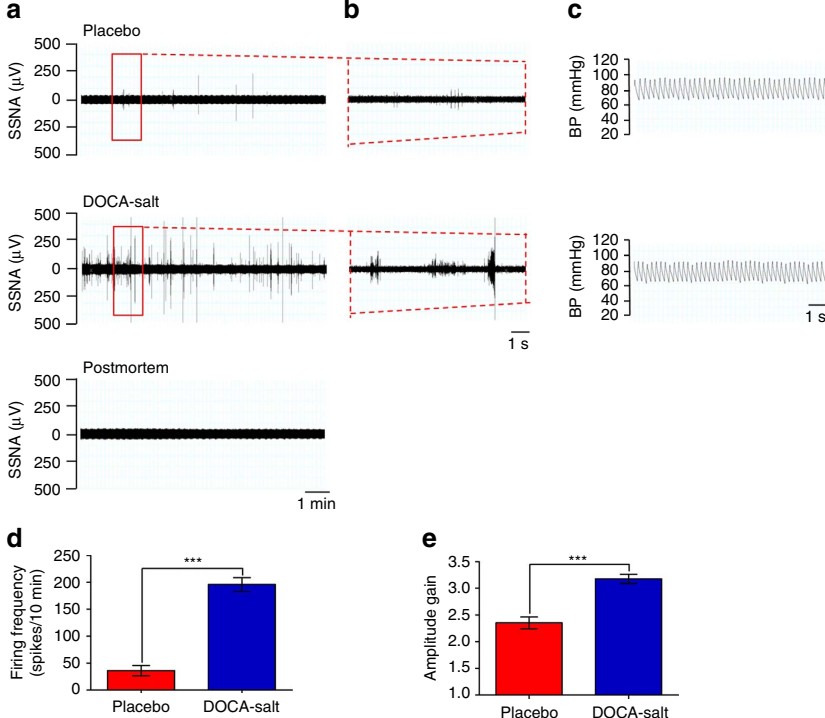

**Figure 2 | Activation of splenic nerve activity is a common response to hypertensive stimuli.** (**a–c**) DOCA-salt hypertensive challenge significantly increased splenic nerve discharge, in a similar way to that observed in AngII-infused mice, as shown by the representative raw signals of SSNA in a time bin of 10 min (**a**) and a particular of 10 s from the red window in **b**. Residual SSNA during postmortem (lower panel in **a**) and BP monitoring (**c**) are shown as well. (**d**) DOCA-salt induced a significant increase in firing frequency, expressed as the mean number of spikes in a time bin, as compared with mice receiving placebo (Placebo $n_{mice} = 6$ and DOCA-salt $n_{mice} = 7$; independent samples Student's $t$-test, $t(11) = -9.654$, ***$P < 0.001$). (**e**) DOCA-salt-treated mice display a significant higher mean amplitude gain of spikes, as compared with mice receiving alone (Placebo $n_{mice} = 6$ and DOCA-salt $n_{mice} = 7$; independent samples Student's $t$-test, $t(11) = -5.977$, ***$P < 0.001$).

(Supplementary Fig. 6b) were significantly inhibited as compared with WT mice. In particular, the adrenergic fibres typically innervating the marginal zone of the spleen, evidenced by the CD169 antigen, were almost absent in α7nAChR KO mice (Supplementary Fig. 6b). Further, AngII challenge in α7nAChR KO mice produced neither the increased blood pressure typically observed in WT mice (Fig. 6a,b) nor the expected T cell egression (Fig. 6c,d). Overall, these data support the existence of a vagus-splenic nerve connection that is recruited by hypertensive stimuli and mediated by α7nAChR.

**Splenic nerve ablation prevents immune activation on AngII.** To define the relationship between SSNA and blood pressure rising in response to hypertensive stimuli, we established a procedure to denervate selectively splenic nerves (SDN) by thermoablation (Fig. 7a). Our first step was to verify that the procedure itself would not damage the artery. Ultrasound Doppler imaging revealed that SDN did not affect the normal pulse wave of the splenic artery (Fig. 7b). Moreover, microCT angiography after splenic denervation showed splenic artery integrity and consequently normal perfusion of the spleen (Fig. 7c,d). To evaluate SDN efficacy, we injected a retrograde neurotracer into the spleen to label coeliac ganglion neurons (Fig. 8a). This procedure confirmed interrupted nerve trafficking between the central nervous system and spleen. In addition, tyrosine hydroxylase innervation was markedly reduced in the splenic arteries of SDN mice, as compared with sham (Fig. 8b), as was splenic noradrenaline (Fig. 8c). To demonstrate the selectivity of the procedure for the splenic nerve, we measured ipsilateral kidney noradrenaline content, which was comparable in SDN and

sham mice (Fig. 8d), thereby affirming there were no confounding off-target SDN effects.

Next, we infused SDN and sham mice with AngII for 28 days using osmotic minipumps to elevate sympathetic nervous system activation in the spleen, as shown by increased tyrosine hydroxylase staining (Supplementary Fig. 7a). SDN mice were protected from elevated sympathetic nervous system activity in the marginal zone of the spleen, delineated by the CD169 antigen (Supplementary Fig. 7a). More importantly, SDN mice were significantly protected from the increased blood pressure induced by chronic AngII that is typically observed in sham mice (Fig. 9a,b; Supplementary Fig. 7b,c). Similar results were obtained with both radiotelemetry (Supplementary Fig. 7b,c) and tail-cuff measurements (Fig. 9a,b). We then asked whether this effect could be due to a modulating function of SSNA on splenic T cells, which are known to be activated by hypertensive challenges, before deployment toward target organs[5–7]. SDN hampered AngII-induced T cell egression from the splenic reservoir, as shown by the total CD3$^+$ T cell content and area in the white pulp (Fig. 9c,d). Indeed, AngII infusion did not reduce T cell content in mice with SDN but did in sham mice (Fig. 9c,d). We also evaluated whether the lack of T cell egression could be caused by failed T cell co-stimulation activation in the absence of an intact splenic nerve. Mice with SDN were protected from AngII-induced CD86 expression (Fig. 9e), the typical hallmark of T cell co-stimulation and hence a key indicator of full activation for target organ colonization in hypertension[6,7]. We next assessed T cell infiltration in target organs by flow cytometry and found that AngII-SDN mice were significantly protected from CD8$^+$ T cell infiltration in the aorta (Fig. 10a) and kidneys (Fig. 10b,c) in terms of both total cells and the number of cells

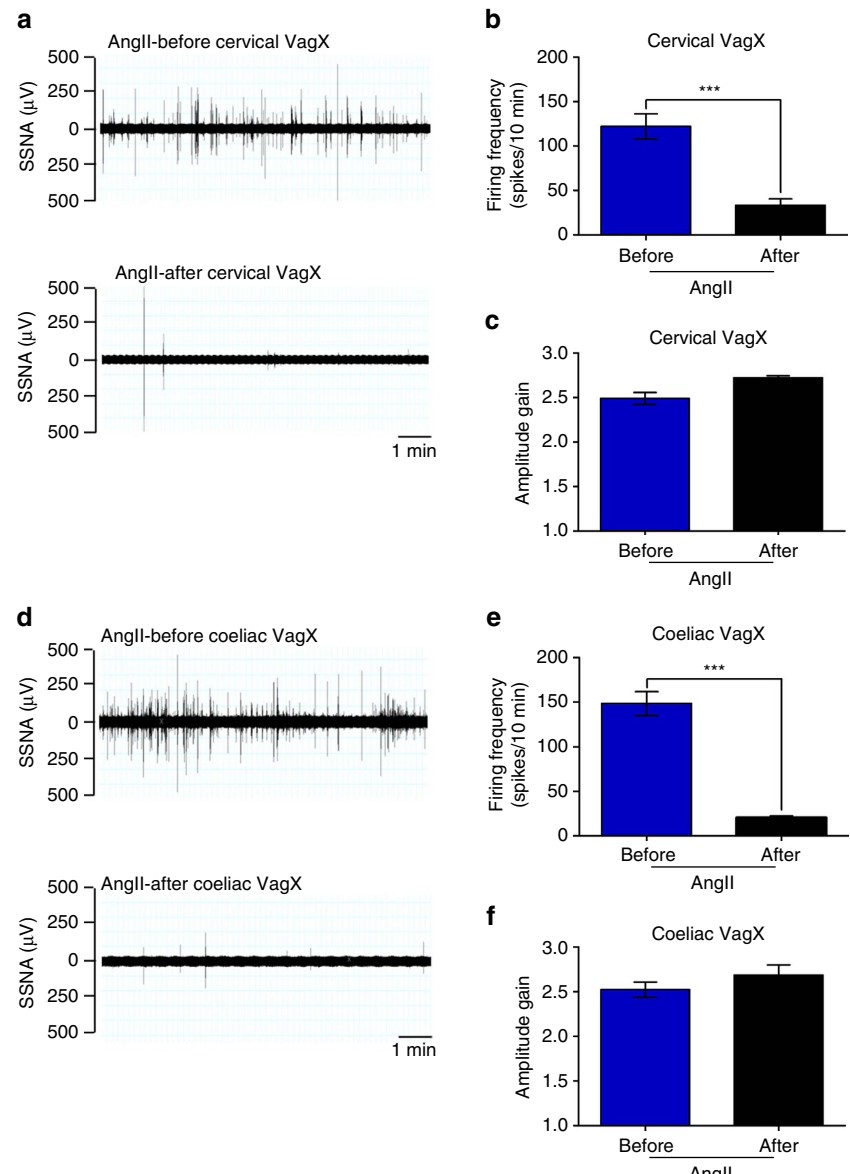

**Figure 3 | Efferent vagus nerve resection blocks the SSNA induced by hypertensive stimuli.** (**a**) Mice infused with AngII were subjected to cervical vagotomy (VagX) while recording SSNA. As shown by the representative raw signals of SSNA, cervical VagX completely abolishes splenic nerve activity. (**b**) Analysis of firing frequency confirmed the inhibitory effect of cervical VagX on AngII-induced SSNA ($n_{mice} = 5$; paired samples Student's $t$-test, $t(4) = 8.524$, ***$P < 0.001$). (**c**) Conversely, no effect of cervical VagX on the mean amplitude gain of spikes was observed, suggesting that this pattern of SSNA was regulated by pathways different from the vagus nerve ($n_{mice} = 5$; paired samples Student's $t$-test, $t(4) = -2.764$, $P = 0.051$). (**d**) To verify whether the effect was due to the afferent or efferent branch of the vagus nerve, a further group of AngII-infused mice underwent coeliac vagotomy (VagX) while recording SSNA. Even in this case, splenic nerve activity was completely inhibited by coeliac VagX. (**e**) Analysis of firing frequency confirmed the inhibitory effect of coeliac VagX on AngII-induced SSNA ($n_{mice} = 5$; paired samples Student's $t$-test, $t(4) = 11.059$, ***$P < 0.001$). (**f**) Yet, no effect of coeliac VagX on the mean amplitude gain of spikes was observed, further supporting that this pattern of SSNA was regulated by pathways different from the vagus nerve ($n_{mice} = 5$; paired samples Student's $t$-test, $t(4) = -1.518$, $P = 0.204$).

primed with markers for homing (CD8+ CD44+ cells) and activation (CD8+ CD69+ cells). Similar results were found for CD4+ T cell infiltrates (Fig. 10a–c). These data were also confirmed by immunohistochemical analyses of the aorta (Supplementary Fig. 8a–d) and kidneys (Supplementary Fig. 9a–d).

## Discussion

Our results demonstrate that hypertensive challenges exploit a cholinergic-sympathetic drive, realized through a vagus-splenic nerve connection, to activate the T cells that eventually migrate to target organs and contribute to blood pressure regulation. We show that stimuli like chronic AngII and DOCA, which are sensed by the circumventricular organs in the brain to induce hypertension[19,23], activate potent splenic nerve discharge. We also report that SSNA depends on an intact vagus nerve efferent that terminates in the coeliac plexus ganglia where the catecholaminergic fibres of the splenic nerve originate. We further demonstrated that the vagal-coeliac-splenic nerve connection regulating blood pressure is mediated by α7nAChR, similar to dynamics previously described in endotoxemia[26].

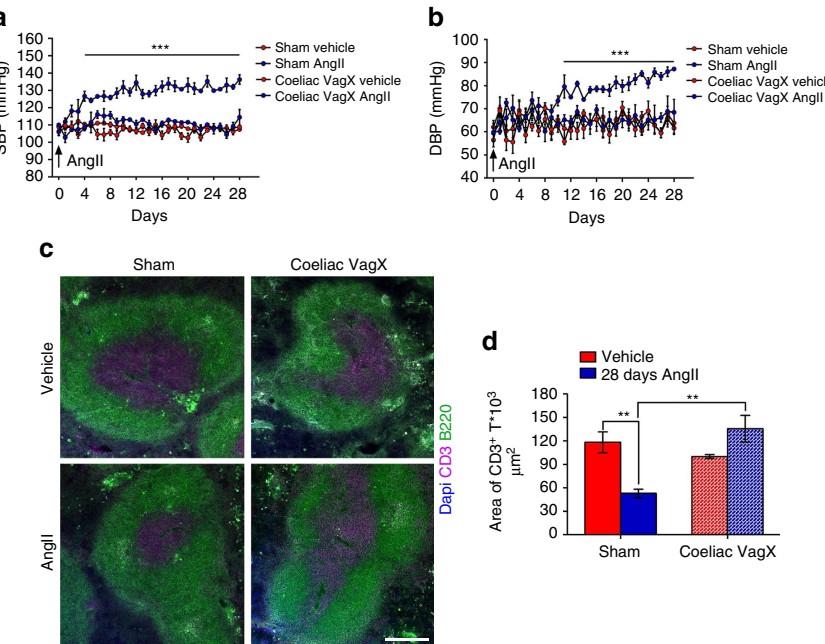

**Figure 4 | Coeliac vagotomy protects from AngII-induced blood pressure increase and T cell egression.** (**a,b**) Mice with left coeliac VagX did not become hypertensive in response to chronic AngII infusion ($n_{mice} = 5$ for each group; two-way ANOVA for repeated measures; (**a**) systolic blood pressure SBP, $F_{(interaction)} = 3.321$, ***$P < 0.001$; (**b**) Diastolic blood pressure DBP, $F_{(interaction)} = 2.284$, ***$P < 0.001$). (**c**) Left coeliac VagX was also effective in inhibiting the T cell egression induced by AngII, as evidenced by the area of $CD3^+$ cells (magenta), representing the white pulp and delimited by $B220^+$ cells (green) delineating the red pulp (scale bar, 200 μm). (**d**) Graph showing the relative quantitative analysis ($n_{mice} = 5$ for each group; two-way ANOVA, $F_{(interaction)} = 20.160$, **$P < 0.01$).

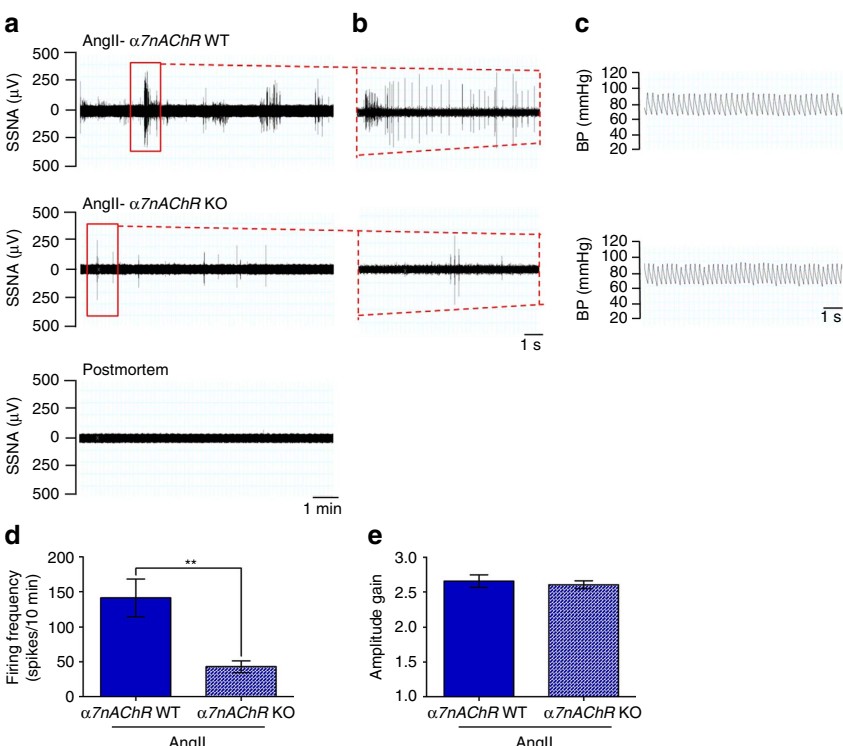

**Figure 5 | α7nAChR is necessary to the activation of SSNA on hypertensive challenges.** (**a–c**) The representative raw signals of SSNA in a time bin (**a**) and in an enlarged particular of 10 s (**b**) show that α7nAChR KO mice have a reduced response to AngII infusion, as compared with WT controls. Residual SSNA during postmortem (lower panel in **a**) and blood pressure monitoring (**c**) are shown as well. (**d**) Analysis of firing frequency confirmed the reduced activity of α7nAChR KO mice on AngII infusion (WT and α7nAChR KO $n_{mice} = 7$; independent samples Student's $t$-test, $t(12) = 3.493$, **$P < 0.01$). (**e**) Accordingly to what observed in VagX, the mean amplitude gain of spikes induced by AngII was unaffected in α7nAChR KO, showing levels comparable to that of WT mice (WT and α7nAChR KO $n_{mice} = 7$; independent samples Student's $t$-test, $t(12) = 0.572$, $P = 0.578$).

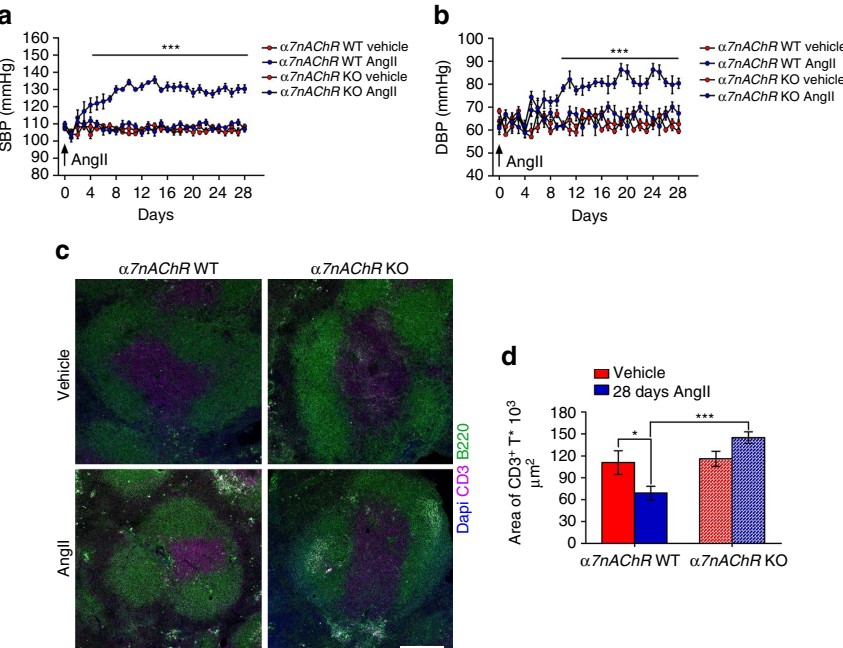

**Figure 6 | α7nAChR KO mice are protected from AngII-induced hypertension and T cell egression. (a,b)** α7nAChR KO mice were protected from hypertension induced by chronic AngII infusion ($n_{mice} = 8$ for each group; two-way ANOVA for repeated measures; **(a)** SBP, $F_{(interaction)} = 5.331$, ***$P < 0.001$; **(b)** DBP, $F_{(interaction)} = 5.087$, ***$P < 0.001$). **(c)** Ablation of α7nAChR was effective in blocking T cell egression on AngII, similarly to vagotomized mice, as evidenced by the area of white pulp, labelled by CD3+ cells (magenta), delimited by B220+ cells (green) of the red pulp (scale bar, 200 μm). **(d)** Graph showing the relative quantitative analysis ($n_{mice} = 7$ WT vehicle; 8 WT AngII; 6 α7nAChR KO Vehicle; 6 α7nAChR KO AngII; two-way ANOVA, $F_{(interaction)} = 8.939$, *$P < 0.05$ and ***$P < 0.001$).

Taken together, our findings—particularly our selective denervation of the splenic nerve—have important translational implications. Mice with splenic denervation were protected from hypertension through impeded T cell activation, egression from the spleen and consequent infiltration of target organs. These results provide a rationale for investigating novel therapeutic strategies based on splenic denervation to treat hypertension.

Historically recognized as a crucial player in hypertension, the sympathetic overdrive promotes the initial blood pressure surge in the early clinical stages of the disease and helps maintain elevated blood pressure levels[3,10]. Moreover, constant adrenergic overactivation is known to contribute, over time, to end-organ damage caused by chronic hypertension and metabolic abnormalities often observed in hypertensive patients. Researchers have sought to understand the mechanisms and effects of renal sympathetic nerves in hypertension[31]. Indeed, it is now widely accepted that, under hypertensive conditions, this system changes in pathological ways that contribute to alterations in sodium reabsorption and thus negatively influence blood pressure control[3,10,31]. Yet the results of clinical trials on renal denervation remain controversial[11–13], lending support to the hypothesis that there may be other mechanisms contributing to the impact of constant sympathetic overactivation in hypertension. Hence, we sought to pursue selective splenic denervation to decipher the immune system's role in hypertension. Although a simple neurectomy could in principle prove the same concepts and provide results, selective denervation obtained by thermoablation has clinical potential for patients for whom the renal denervation has failed. Our results strongly suggest splenic denervation may be a useful tool in treating resistant hypertension in humans.

Abundant research has confirmed that the immune system plays a key role in hypertension. Yet the process by which a hypertensive challenge signals the immune system has remained largely unclear. Intriguingly, even as researchers in the field of hypertension began dissecting the role of the immune system, immunologists of the same period discovered that neural circuits can modulate immune cells[15,32]. Meanwhile, decades of studies focusing on the neurophysiological basis and broader effects of the inflammatory reflex converged on the spleen. Action potentials are transmitted from the vagus nerve to the coeliac ganglion, where the splenic nerve originates, and activated adrenergic splenic neurons can regulate a specific T cell subset in splenic white pulp. In the spleen, we now know, neural signals deal with immunity directly[33,34]. We are beginning to understand the complex relationship between the nervous system and immunity.

As researchers continue to decipher these connections, nervous circuits' largely undiscussed role in blood pressure regulation[10] will become increasingly important. On the basis of our results, it is tempting to speculate that sympathetic overactivity in hypertension has effects beyond the kidney and baroreflexes. The novel concept that the autonomic nervous system can have long-term effects on cardiovascular pathologies through immune system regulation requires us to significantly rethink the role of the autonomic nervous system[14]. Interestingly, it is well known that immune organs, and particularly the spleen, are directly innervated by the sympathetic nervous system[7,35]. In addition, the autonomic system may be a powerful regulator of immunity: although AngII has been shown to be responsible for a variety of actions that may contribute to the development of hypertension, several milestone papers demonstrate that AngII also activates the peripheral sympathetic nervous system and affects immune responses[19]. Intracerebral ventricular AngII infusion increases central sympathetic nerve activation and enhances the immune response in the periphery[36]. These findings are particularly important because they demonstrate that AngII-induced effects on both hypertension and immune system activation are not

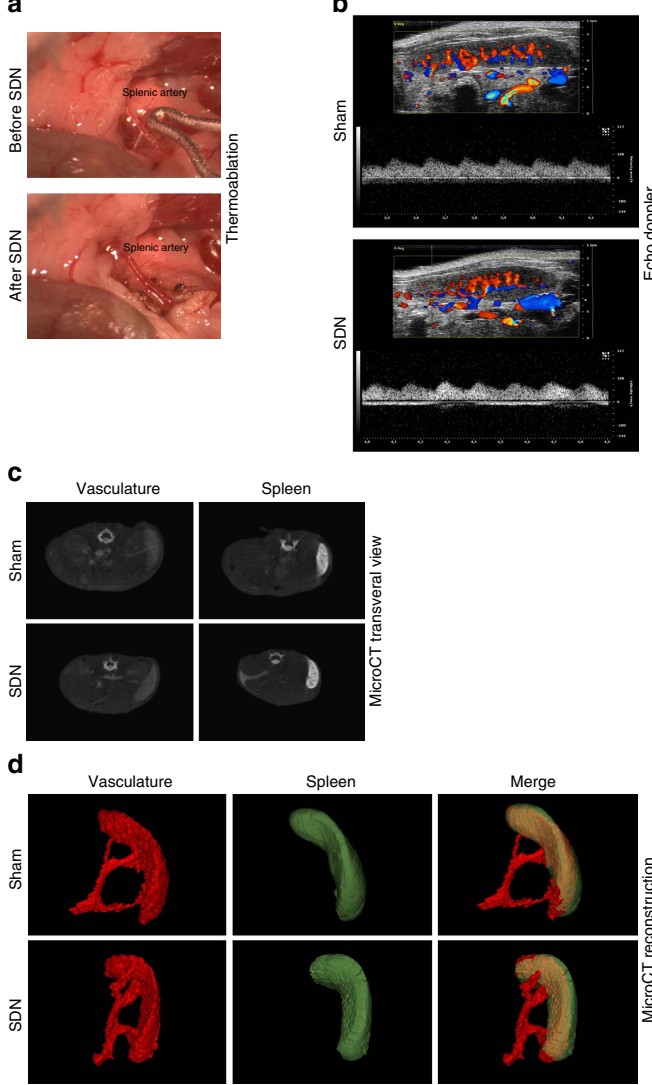

**Figure 7 | Establishment of a procedure for selective splenic denervation.**
(**a**) Example of surgical procedure for selective SDN, showing the splenic artery before (upper panel) and after (lower panel) denervation in a representative animal. (**b**) Ultrasound Doppler imaging showing a normal pulse wave of the splenic artery in splenic denervated mice, comparable to that of sham. Colour Doppler also indicate a normal perfusion of the spleen after denervation. (**c**) Anatomical transversal reconstruction of microCT angiography evidencing of correct uptake of the contrast agent in the spleen. (**d**) 3D reconstruction demonstrating integrity of splenic artery after denervation and consequent normal perfusion of the spleen.

caused by direct AngII influence on the vasculature and immune cells, thus suggesting that central signals may be integrated by the immune system to drive hypertension.

We believe the results presented in this paper are significant because they reveal, for the first time, a previously unknown sympathetic pathway in hypertension. The brain-to-spleen connection's realization through a cholinergic-sympathetic nervous drive resembles the cholinergic anti-inflammatory pathway that is protective when activated by electrically stimulating the vagus efferent during endotoxemia[24–27]. So far, this pathway has been primarily considered a way for the nervous system to modulate immune response to bacterial toxins. More specifically, vagus stimulation during septic shock determines the coupling of cholinergic pre-ganglionic neurons

and postganglionic sympathetic nerves, linkages that are ultimately responsible for attenuating splenic TNFα production[24–27].

α7nAChR has been well characterized as the molecular mediator of the cholinergic anti-inflammatory pathway and therefore integral to the vagus nerve/sympathetic drive in the splenic district[26,34]. Indeed, α7nAChR can be found in several tissues, including the brain, ganglia, immune cells and so on. Our experiments showed that α7nAChR KO mice are protected from AngII-induced hypertension. Because α7nAChR is a demonstrable mediator of the cholinergic anti-inflammatory pathway expressed in cytokine-producing splenic macrophages[25,34] but also integrates the sympathetic and parasympathetic systems in presynaptic neurons of the splenic nerve[26], we sought to understand where α7nAChR acts in hypertension. That *α7nAChR* KO mice had markedly inhibited AngII-elicited SSNA supported the idea that α7nAChR plays a role in mediating autonomic integration among parasympathetic and sympathetic drives in the coeliac ganglion (upstream from the splenic sympathetic drive). Interestingly, vagus nerve stimulation was recently shown to mediate protection from kidney ischemia-reperfusion injury through α7nAChR in splenocytes[37]. Lastly, our findings highlight the possibility that cholinergic anti-inflammatory pathway activation during endotoxemia may be also hemodynamic and therefore relevant to fighting the hypotensive shock that is one of the fatal complications of sepsis.

## Methods

**Animals and drugs.** All animal handling and experimental procedures were performed according to European Community guidelines (EC Council Directive 2010/63) and the Italian legislation on animal experimentation (Decreto Legislativo D.Lgs 26/2014). All efforts were made to minimize suffering, and the principles of Replacement, Reduction and Refinement (that is, the 'three Rs') were applied to all experiments.

C57Bl/6J male mice, aged 8-12 weeks, were purchased from Jackson Laboratory and used in all experiments. *Chrna7* (α7nAChR) knockout and WT littermates (stock number 003232, Jackson Laboratory) were used where indicated. Mice were housed in an air-conditioned room (temperature $21 \pm 1\,°C$, relative humidity $60 \pm 10\%$), with lights on from 06:00 to 18:00, and had sawdust as bedding, pellet food and tap water *ad libitum*. After surgical procedures, mice were housed in recovery boxes (temperature $37\,°C$) and carefully monitored for several days.

In acute treatments administered during electrophysiological recordings, mice received Sodium Nitroprusside (SNP; Sigma Aldrich), Phenylephrine (Phe; Sigma Aldrich) and Hexamethonium bromide (Sigma Aldrich) at the dosage indicated in the appropriate section.

A cohort of mice received $0.5\,mg\,kg^{-1}\,day^{-1}$ of AngiotensinII (AngII; Sigma Aldrich) or Vehicle (NaCl 0.9%), delivered subcutaneously with osmotic minipumps (model 2004, ALZET)[7]. Another group of mice underwent subcutaneous implantation of a 50-mg pellet of DOCA or placebo as control (Innovative Research of America) without uninephrectomy[23]. Mice were maintained on standard chow with *ad libitum* access to tap water and $0.15\,mol\,l^{-1}$ (0.9%) NaCl.

**Electrophysiological recording.** In the first set of experiments, SSNA was recorded in mice at maximum blood pressure response to either sodium nitroprusside or phenylephrine, and in the next set of experiments, SSNA was recorded three days after receiving either AngII or DOCA-salt. Mice infused with vehicle or implanted with a placebo pellet were used as respective controls. Mice were anaesthetized with 5% isoflurane and subsequently maintained with 1.5–2% (supplemented with $1\,l\,min^{-1}$ oxygen). Blood pressure was monitored during the entire experiment with a single-pressure catheter (Millar, SPR-100) inserted in the left femoral artery and connected to a pressure transducer (Millar, MPVS ULTRA)[7]. Body temperature was maintained between 37 and $38\,°C$ by a homoeothermic blanket.

An abdominal incision was made and the intestinal tract was carefully moved aside to expose the splenic artery. First, the splenic artery was isolated, and then the splenic nerve was carefully separated from surrounding tissue. A bipolar stainless steel electrode (MLA1214 Spring Clip Electrodes, ADInstruments) was adjusted to the nerve size and gently placed on the splenic nerve, according to previously described procedures[20–22,38]. The ground wire was plugged into mouse soft tissues. When optimum recording was obtained, the electrode was covered with silicone gel. SSNA was recorded from the implanted electrode continuously during the next

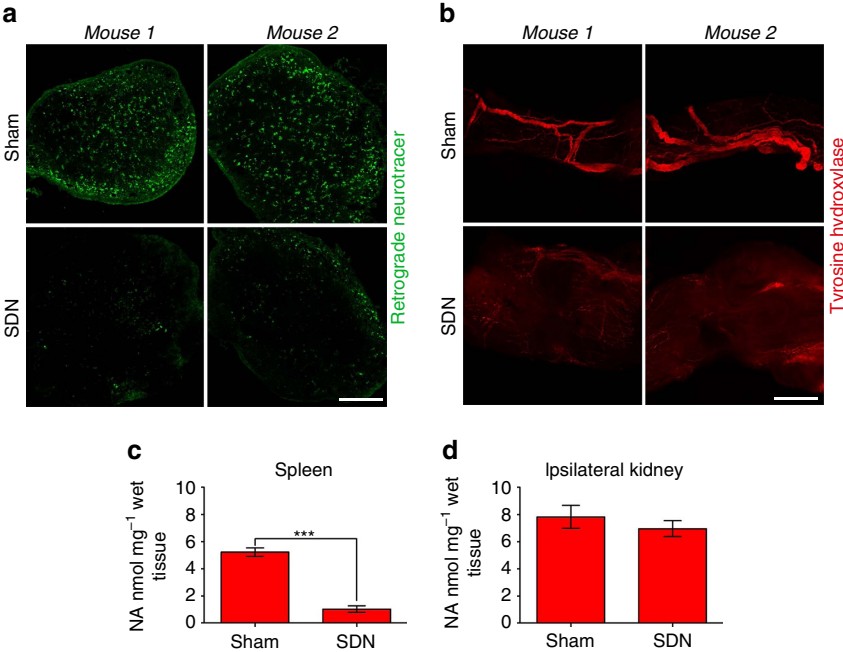

**Figure 8 | Selective splenic denervation blocks sympathetic nervous system in the spleen.** (**a**) Coeliac ganglion neurons, labelled with a retrograde neurotracer injected in the splenic parenchyma, are significantly reduced in SDN mice as compared with sham controls. ($n_{mice} = 6$ for each group; scale bar, 200 μm). (**b**) Significant reduction of tyrosine hydroxylase immunofluorescence in splenic artery of two representative SDN as compared with Sham mice ($n_{mice} = 6$ for each group; scale bar, 50 μm). (**c**) Noradrenaline levels were significantly reduced in the spleen of SDN mice as compared with Sham mice (Sham and SDN groups $n_{mice} = 8$; independent samples Student's $t$-test, $t(14) = 10.606$, ***$P < 0.001$). (**d**) Analysis of noradrenaline content in the ipsilateral kidney was unaltered in SDN mice as compared with Sham, revealing no off-target effect of the procedure (Sham and SDN groups $n_{mice} = 8$; independent samples Student's $t$-test, $t(14) = 0.853$, $P = 0.408$).

2 h, as was arterial blood pressure. Mice were then killed by isoflurane overdose, and SSNA was recorded for a further 30 minutes to estimate postmortem residual activity.

The splenic nervous signal was amplified (gain × 10,000) and then sampled at 4 kHz with a digital amplifier (Animal Bio Amp; ADInstruments). The raw signal was filtered at 300–1,000 Hz and expressed as μV. Background noise estimates for the nerve activity trace were based on postmortem recording. Raw splenic nerve activity and blood pressure signals were monitored using a PC that was online during the entire recording. Operators were blinded to the experimental group during recording.

Data were collected using a Power Lab data acquisition system and analyzed with Lab Chart 7 (Spike Analysis Module). Typically, a sympathetic burst was identified as a signal above the threshold determined by the background noise of the postmortem recording. In acute experiments, SSNA counting was measured using a bin time of 5 min at the plateau of blood pressure response after a bolus of sodium nitroprusside ($2.5 μg g^{-1}$ of body weight in a volume of 25 μl of saline followed by 50 μl of saline) or phenylephrine ($20 μg g^{-1}$ of body weight in a volume of 25 μl of saline followed by 50 μl of saline)[20]. Background noise was recorded after ganglionic blockade with hexamethonium[20] ($50 μg g^{-1}$ of body weight in 25 μl) and during postmortem. In chronic experiments, SSNA counting was performed using a bin time of 10 min and collected every 20 min. Firing frequency was measured as the mean value of total number of spikes in a time bin. The amplitude gain was calculated as the ratio between mean spike amplitude and spike identification threshold in a time bin.

**Vagotomy.** Surgical cervical unilateral vagotomy was performed via a cervical midline incision that exposed the left vagus trunk, which was cut with forceps. Coeliac vagotomy was performed with the same procedure, cutting the distal end of the coeliac branch of the vagus nerve. In a first set of experiments, both procedures were executed while recording SSNA. Vagus nerve was exposed while preparing the mouse for the electrophysiological recording. After the acquisition of two SSNA time bins, the cervical or coeliac vagus nerve was transected, while SSNA recording continued for at least two additionally time bins. In a second group of mice, coeliac vagotomy was performed and AngII minipumps implanted subcutaneously for chronic experiments.

**Splenic denervation.** After anaesthesia with isoflurane (2–5 Vol%), supplemented with $1 \, l \, mi^{-1}$n oxygen, the mouse abdominal cavity was opened and the splenic artery was carefully exposed. The selective SDN was performed by thermoablation applied with a thermal cautery (GEIGER Medical Technologies). The cautery was gently placed on the splenic artery for 5–6 s until the splenic artery was dilated. In sham-operated mice, the abdomen was opened and the splenic artery was exposed, but thermoablation was not performed.

**Neurotracer injection.** The efficacy of splenic nerve ablation was evaluated by neuronal labelling, obtained by retrograde transport, in coeliac ganglion. Mice were anaesthetized with isoflurane (2–5 Vol%), the spleen was exposed through a left flank incision and 10 injections of 2 μl hydroxystilbamidine, methanesulfonate 2% (Molecular Probes, H22845) were injected into the parenchyma[7]. Mice were allowed to recover, and seven days later they were deeply anaesthetized so that coeliac ganglia from each side could be collected.

**Blood pressure measurements.** Arterial blood pressure was monitored by radiotelemetry where indicated. HD-X11 pressure transmitters (Data Sciences International) were implanted in mice anaesthetized with isoflurane. The pressure-sensing catheter was inserted into the aortic arch, and the transmitter body was placed in a subcutaneous pouch on the back. Following surgery, the mice were allowed to recover for at least 1 week. Radio signals from the implanted transmitter were captured by the Physiotel RPC-1 receiver (Data Sciences International), and the data were stored online using the Dataquest Ponemah 4.9 data acquisition system (Data Sciences International) according to standardized procedures[7,39,40]. BP was recorded daily for 2 h (10am-12pm) to monitor the effects of AngII.

Noninvasive blood pressure measurement was performed by tail-cuff plethysmography (BP-2000 Series II, Visitech Systems) in conscious mice daily for 2 hours (10 am–12 pm)[7,39,40]. Operators were blinded to the experimental group during blood pressure monitoring.

***In vivo* imaging.** Ultrasound analysis was performed with Vevo 2100 (VisualSonics, Toronto, Canada) equipped with 40MHz transducer, as previously described[7]. Mice were anaesthetized with isoflurane anaesthesia system (5 Vol% induction, 1.5 Vol% maintenance supplemented with $1 \, l \, min^{-1}$ oxygen) and taped to a warmed bed, positioned on the right side. The spleen area was shaved, and the probe was positioned under the chest to get a longitudinal view of the spleen. To visualize splenic artery perfusion and function after splenic denervation, PW Doppler was applied in order to measure peak flow velocity. Colour Doppler was applied to images of the spleen to verify organ perfusion after splenic denervation.

MicroCT scanning was performed with SKYSCAN 1178 (SKYSCAN, Kontich, Belgium), as previously described[7]. The tube parameters were set to 50 kV and 615

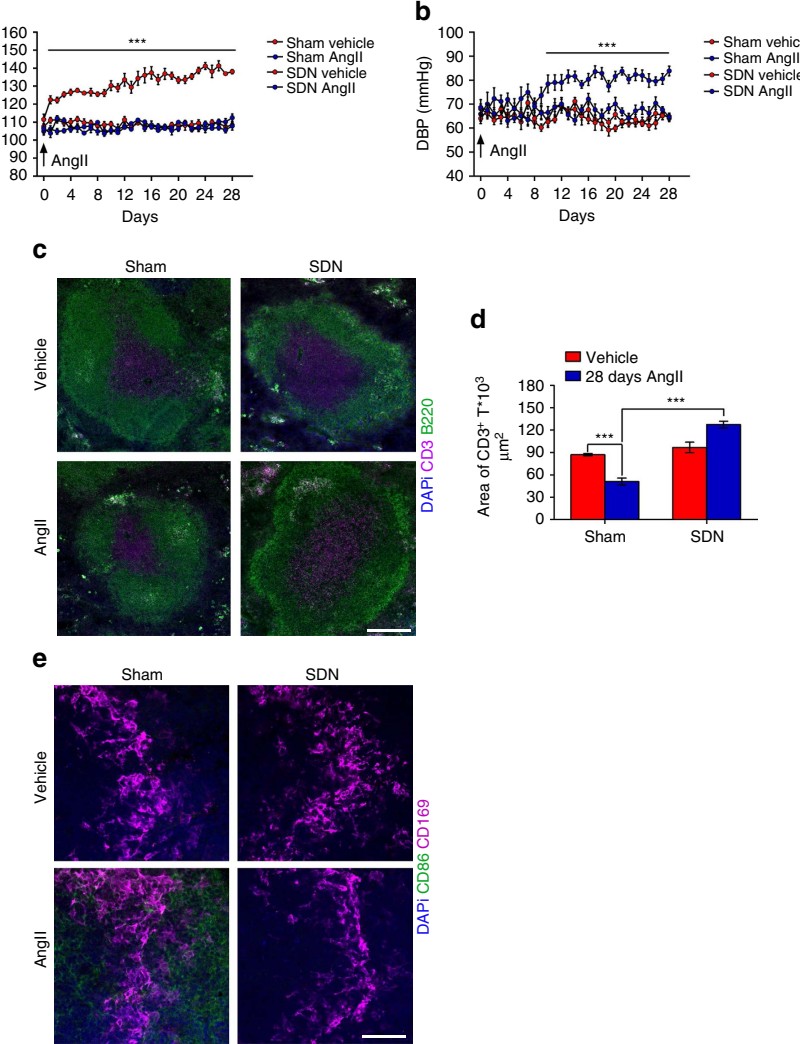

**Figure 9 | Splenic denervation protects from hypertension and T cell egression from spleen. (a,b)** Mice with splenic denervation were protected from AngII-induced hypertension, as compared with Sham mice and SDN mice with vehicle alone (Sham-vehicle, Sham-AngII and SDN-vehicle $n_{mice} = 10$; SDN-AngII $n_{mice} = 11$; two-way ANOVA for repeated measures; **(a)** SBP, $F_{(interaction)} = 6.723$, \*\*\*$P < 0.001$; **(b)** DBP, $F_{(interaction)} = 3.070$, \*\*\*$P < 0.001$). **(c)** Mice with splenic denervation are protected from the T cell egression induced by AngII, as evidenced by the area of CD3$^+$ cells (magenta), recognizing white pulp and delimited by B220$^+$ cells (green) delineating red pulp ($n_{mice} = 5$ for each group; scale bar, 200 μm). **(d)** Graph showing the relative quantitative analysis ($n_{mice} = 5$ for each group; two-way ANOVA, $F_{(interaction)} = 46.384$, \*\*\*$P < 0.001$). **(e)** AngII fails to induce co-stimulation of T cells in mice with splenic denervation, as evidenced by reduced CD86$^+$ cells (green) expression in the marginal zone of the spleen, labelled by CD169$^+$ cells (magenta) ($n_{mice} = 5$ for each group; scale bar, 50 μm).

μA, exposure 480 ms, with a rotation step of 0.360°. Mice were anaesthetized with ketamine-xylazine and positioned on the bed. Two different scans were performed: the first scan occurred immediately after contrast agent injection (ExiTron nano 12,000, 100 μl per mouse) to determine vascular perfusion; after 24 h, the second scan was performed to obtain a spleen tissue image using contrast agent accumulation in the spleen. Spleen reconstruction was performed with 3D Slicer[41]. In both scans, the image was cropped to the spleen region, then the images were co-registered with an affine transformation. The co-registered images were labelled and 3D rendered to highlight the vascularization in the first scan (shown in red) and contrast tissue absorption (shown in green) in the second scan.

**Tissue isolation.** At the end of each experiment, tissues were isolated for subsequent analyses. The splenic artery was isolated and fixed with 4% paraformaldehyde (PFA). Spleen and aorta were collected and embedded in OCT for cryo-sectioning. Kidneys were explanted and embedded in paraffin for immunohistochemistry. Coeliac ganglion was post-fixed in Zamboni's Fixative and embedded in OCT for cryo-sectioning. For biochemical analyses, spleen and kidneys were explanted and flash-frozen in liquid nitrogen.

**Flow cytometry.** After mice were exsanguinated, both kidney and aorta were collected. Kidney cell suspension was obtained by mechanically disrupting two decapsulated kidneys in 10 ml of RPMI 1640 medium (GIBCO, Invitrogen)

supplemented with 5% FBS, which was then passed through a 70 μm sterile filter (Falcon, BD). The resulting cell suspension was centrifuged at 300g for 10 min to pellet the cells. To isolate leukocytes from cell suspension, the pellet was suspended in 36% Percoll (Sigma), gently overlaid onto 72% Percoll and centrifuged at 1,000g for 30 min at RT. Cells were isolated from Percoll interface and washed twice in medium at 300g for 10 min at 4 °C. Aorta cell suspension was obtained by mechanically disrupting the tissue and digesting the suspension in Digestion Cocktail (450 U ml$^{-1}$ Collagenase I, 125 U ml$^{-1}$ Collagense XI, 60 U ml$^{-1}$ Hyaluronidase I-S) for 40 min at 37 °C with gentle vortexing. Cell suspension was then passed through a 70 μm sterile filter (Falcon, BD), and the resulting cell suspension was centrifuged at 300g for 5 min to pellet the cells.

Lymphocytes from both organs' leukocytes were enriched with Mouse T Lymphocyte Enrichment Set-DM (BD IMag), and the number of the cells was assessed using trypan blue and an automated counter (Countess, Life Technologies). Total kidney and aorta lymphocytes were analyzed with flow cytometry. First, samples were pre-incubated with anti CD-16/32 Fc receptor and then incubated with various combinations of mAbs for immunofluorescence staining using BD FACSCanto (BD Biosciences). The fluorochrome-conjugated mAbs to mouse antigens used for flow cytometry analysis were as follows: PerCP-Cy5.5 anti-CD8a (53-6.7), FITC anti-CD4 (RM4-5), APC anti-CD69 (H1.2F3), APC-Cy7 anti-CD45 (30-F11), PE-Cy7 anti-CD44 (IM7) (1:100; BD PharMingen). The data were analyzed using FlowJo Software (V 10.0.8).

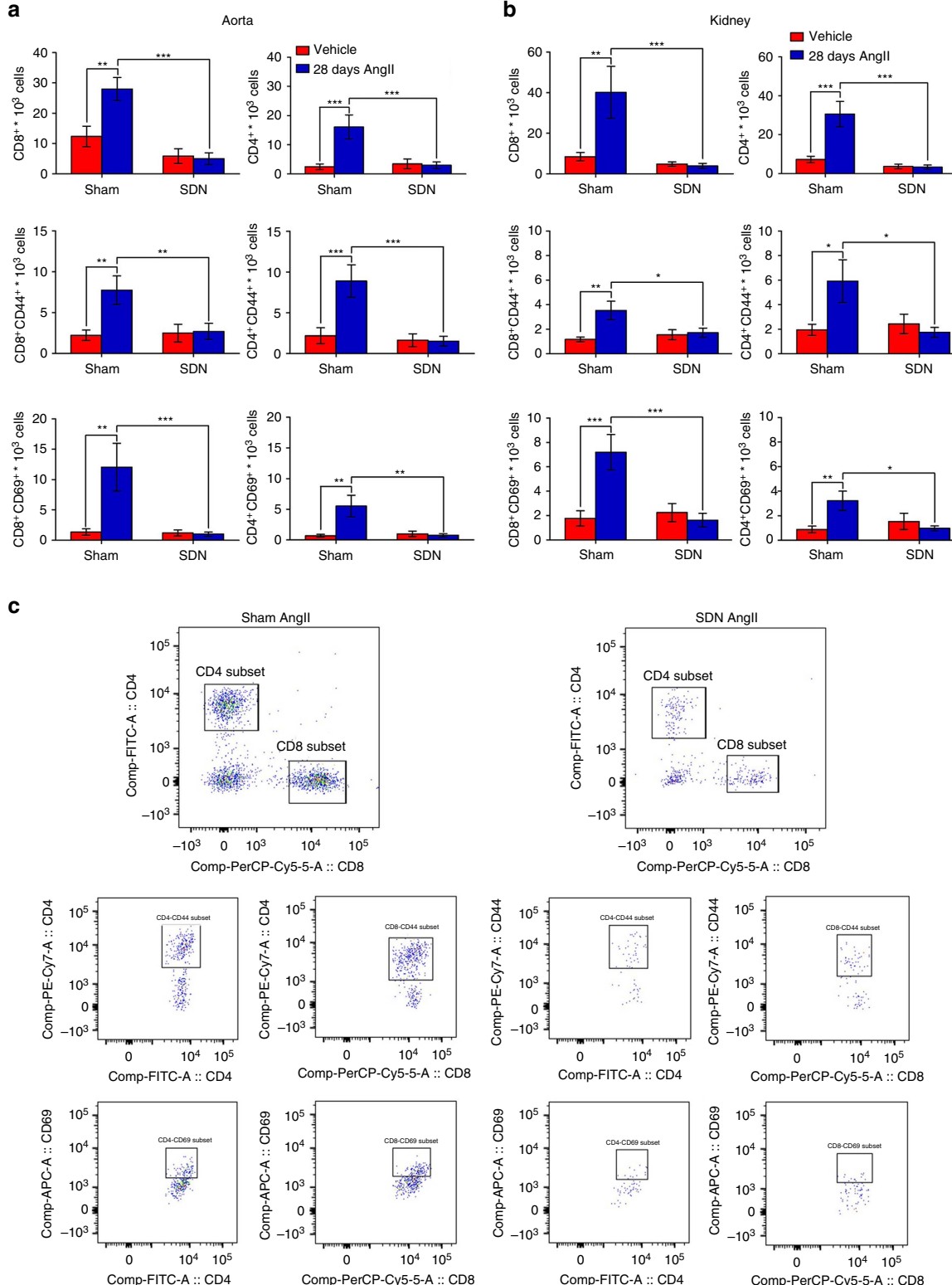

**Figure 10 | Splenic denervation protects from the T cells infiltration in target organs of hypertension.** (**a,b**) Flow cytometry analysis shows that SDN mice were protected from the AngII-induced increase of total CD8$^+$ and CD4$^+$ T cells (upper panels) in aorta (**a**) and (**b**) kidneys. SDN mice displayed also a reduced amount of cells positive for homing (CD44) and activation (CD69) antigens (middle and lower panels, respectively) ($n_{mice} = 9$ for each group). Significance of two-way ANOVA was as follow for aortas: CD8 $F_{(interaction)} = 7.922$, CD8-CD44 $F_{(interaction)} = 5.071$, CD8-CD69 $F_{(interaction)} = 7.344$, CD4 $F_{(interaction)} = 9.122$, CD4-CD44 $F_{(interaction)} = 7.867$, CD4-CD69 $F_{(interaction)} = 8.432$. Significance of Two-way ANOVA was as follow for kidneys: CD8 $F_{(interaction)} = 13.88$, CD8-CD44 $F_{(interaction)} = 5.382$, CD8-CD69 $F_{(interaction)} = 7.202$, CD4 $F_{(interaction)} = 6.248$, CD4-CD44 $F_{(interaction)} = 3.810$, CD4-CD69 $F_{(interaction)} = 10.89$. *$P < 0.05$, **$P < 0.01$ and ***$P < 0.001$. (**c**) Representative plots and gating strategies are shown for kidneys.

**Immunofluorescence analysis.** Splenic arteries used for immunofluorescence analysis were fixed with 4% PFA, passed in PBS for incubation with antibodies and then coverslipped with anti-fading medium (DABCO, Fluka). For coeliac ganglion and spleen, 25 μm sections were obtained with cryostat microtome. Coeliac ganglions were directly mounted and coverslipped for analysis of neurons labelled with retrograde fluorescent neurotracer. Slides from the spleen were post-fixed in PFA (4%) for 15 min and processed for staining.

The following primary antibodies were used: Sheep anti-tyrosine hydroxylase (1:800; AB1542, Millipore); Rat anti-CD169 (1:200; MCA884, Serotec); Hamster anti-CD3 (1:100; MCA269OT, Serotec); Rat anti-CD45R/B220 (1:50; 550286, BD Pharmigen); Rabbit anti-CD86 (1:100; NB110-55488, Novus Biologicals). Sections were incubated with their respective secondary antibodies conjugated to Alexa Fluor 488 or Cy3 (1:200; Jackson Immunoresearch). Slides were then coverslipped with DAPI-containing medium (Vector).

All coverslipped, mounted tissue sections were scanned using a Zeiss 780 confocal laser-scanning microscope, as previously described[7]. A 405 Diode laser was used to excite DAPI; a 488 nm argon laser to excite Alexa Fluor 488 and a 543 HeNe to excite Cy3. Quantitative analyses were determined using Image J software (NIH).

**Immunohistochemistry analysis.** Kidney sections were deparaffinized and rehydrated before undergoing antigen retrieval. Aortas, embedded in OCT, were post-fixed with PFA (4%) for 15 minutes. Slides were processed with the primary antibodies anti-CD8 (1:50; 550286, BD PharMingen) and anti-CD4 (1:50; 550280, BD PharMingen). Samples were incubated with biotinylated secondary antibodies (1:200; Vector) and then processed with DAB (Vector). Hematoxylin (Sigma Aldrich) was used for counterstaining. The number of $CD8^+$ and $CD4^+$ cells per $\mu m^2$ was determined using the Image J software Cell Counter plugin analysis tool (NIH). All images were captured using a DMI3000B Leica optical microscope provided by Leica Cameras (Leica Microsystems) and processed with the Leica Application Suite (LAS V3.3).

**ELISA for noradrenaline assay.** Noradrenaline levels were measured in duplicate for each spleen and kidney sample. Samples were extracted with a buffer containing 0.1 HCl and 1 mM EDTA and assayed using a high-sensitivity ELISA kit (RE59261, IBL International), following the manufacturer's instructions. Results are expressed as nmol mg$^{-1}$ of wet tissue.

**Statistical analysis.** All the experiments were replicated within the laboratory. Sample size was pre-estimated from the previously published research and from pilot experiments performed in the laboratory. Data are presented as mean ± s.e.m. Data distribution was assessed with the Shapiro–Wilk normality test and D'Agostino Pearson test, and assumption of homogeneity of variance was tested using Levene's test of equality of variances. For amplitude gain analysis, unequal variance between groups was observed in a minority of cases, and a Welch correction was performed for all comparisons. Statistical significance was assessed with the appropriate test according to each experimental design, as detailed in figure legends. After confirming that all data had normal distributions, we applied Student t-test for either independent samples or paired samples, according to the experimental design and as specified in the figure legends. Multiple group analysis was performed with two-way ANOVA followed by Bonferroni's post hoc. Analysis for repeated measures was applied when required by the experimental setting. $P < 0.05$ was considered significant. Statistical analyses were performed with SPSS 23.0 (IBM Software) and graphs were made with GraphPad Software PRISM5.

**Data availability.** The authors declare that all other data supporting the findings of this study are available within the article and its Supplementary Information files and from the corresponding authors upon request.

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

## Acknowledgements

This work was supported by the following grants: PON01_00829 and PON01_01227 by MIUR (Italian Ministry of University and Research) to G.L.; 'Ricerca corrente' by MOH (Italian Ministry of Health) to D.C. and and Pasteur Institute—Fondazione Cenci Bolognetti to D.C.

## Author contributions

D.C. and G.L. conceived the research, supervised experiments, analysed data, performed statistical analysis and wrote the manuscript. M.P., F.P., V.F., R.I., S.F., L.C., R.C., M.D.L. and G.C. performed experiments. M.P. and L.C. performed experiments, collected, analysed and discussed data.

## Additional information

**Competing financial interests:** The authors declare no competing financial interests.

**How to cite this article**: Carnevale, D. *et al.* A cholinergic-sympathetic pathway primes immunity in hypertension and mediates brain-to-spleen communication. *Nat. Commun.* **7,** 13035 doi: 10.1038/ncomms13035 (2016).

