## [Peer Review File · Nature Communications]

Reviewers' comments:

Reviewer #1 (expert in hypertension, sympathetic nervous system)

Remarks to the Author:

The manuscript by Carnevale D et al. show that selective splenic denervation protect against AngII-induced hypertension though the inhibition of both CD3+T cell egression and CD86 T cell expression in the spleen, and results in the inhibition of CD8+ T cell infiltration in the aorta and kidney. The results of vagotomy and $\alpha 7$ nAChR-KO suggest the complex relationships between the nervous system and immunity. The manuscript by Carnevale D et al. show that selective splenic denervation protect against AngII-induced hypertension though the inhibition of both CD3+T cell egression and CD86 T cell expression in the spleen, and results in the inhibition of CD8+ T cell infiltration in the aorta and kidney. The results of vagotomy and $\alpha 7$ nAChR-KO suggest the complex relationships between the nervous system and immunity. This is a well-written paper containing interesting results which merit publication. However, a number of points need clarifying and certain experiments require further justification. These are given below.

1. Figure 3 show that vagotomy inhibited Ang II-induced splenic nerve activity (SSNA). Can vagotomy inhibit Ang II-induced hypertension and T cell egression?
2. Figure 4 show that $\alpha 7$ nAChR-KO inhibited Ang II-induced SSNA and hypertension. Can $\alpha 7$ nAChR-KO inhibit Ang II-induced T cell egression?

Reviewer #2 (expert in neural control of the spleen)

Remarks to the Author:

The results show that experimental hypertension in murine models are associated with early splenic nerve discharges as measured by recording compound action potentials. These responses are modulated after vagotomy. Hypertension fails to develop in alpha-7nAChR KO animals. The authors conclude that selective splenic denervation protects against the onset of hypertension, and suggest that this is a new insight into the role and mechanism in resistant hypertension.

The findings are highly novel, timely, and should be of widespread interest to the fields of hypertension, immunology, vascular biology and neuroscience.

The approaches are valid. The data and presentation quality are extremely high. Statistical approaches are appropriate.

The data support the conclusions.

Minor comments:

1. The authors note that others "have demonstrated the
19 existence of the so-called cholinergic anti-inflammatory reflex that is mediated by the celiac branch
of the
20 vagus nerve and specifically targets splenic innervation during septic challenges^{21,22}." It would be
important to include the recent JCI paper (Vagus nerve stimulation mediates protection from kidney
ischemia-reperfusion injury through $\alpha 7$ nAChR+ splenocytes <https://www.jci.org/articles/view/83658>).
This a strong argument that vagus to spleen, as defined by the inflammatory reflex, and you also

propose, significant and functional. It has been challenged by other groups because it is so novel, so explanation here is important.

2. The authors note that " AngII challenge in alpha7nAChR KO mice did not produce the increase in blood pressure typically observed in WT mice (Fig. 4g,h). Overall, these data support the existence of a vagus-splenic nerve connection that is recruited by hypertensive stimuli and mediated by alpha7nAChR." Additional explanation here is needed. Do the authors have evidence that the requirement for alpha7nAChR is in the brain? or in the ganglia? In the spleen, the acetylcholine source is from lymphocytes, so how does this affect blood pressure?

3. The abstract as written addresses only a small component of the major findings....I recommend adding additional results to the abstract.

Reviewer #3 (expert in neural control of the spleen)

Remarks to the Author:

This is an innovative and original article showing that a cholinergic-sympathetic pathway mediated by the alpha7Receptor and the activation of splenic T cells prime immunity during hypertension.

However, there are some considerations that should be clarified.

1. The most innovative results of the article are the neuromodulation of the immune system to control hypertension through the T cells. The results correlate the AngII treatment with the splenic egression of the lymphocytes and infiltration of the CD8 T cell in the aorta. The authors are very convincing showing that those mechanisms are through the splenic nerve as demonstrated by the SDN. However, the study does not show how much the splenic egression of the T cells and the aorta infiltration contributes to Angiotensin-induced hypertension. Is this an effect contributing to hypertension too or it is just modulating immunity during hypertension? The authors should consider performing an AngII treatment in nude mice (without T cells) to determine whether T cells are required for (or contribute to) the AngII-induced hypertension.

2. The authors have a bias referencing previous studies. Page 5, lanes 12 and 20, cite references 21 and 22 for the cholinergic- sympathetic control of the spleen. However, ref 22 does not show anything related to the spleen. Instead, the authors appear to ignore critical references from other labs. For instance, the cholinergic control of sympathetic innervation of the spleen modulating inflammation was first reported by Huston et al (2006), J. Exp. Med. 203(7):1623-8 and confirmed later by Refs 21 and Ref 25 Vida et al. (2011) J Immunol 186, 4340-4346. Likewise, Page 10, lane 8-9 cited ref 29, for the role of T cells in the cholinergic-splenic nerve regulation of immunity, but this mechanism was first reported by Peña et al (2011) J Immunol 187(2):718-25.

3. In general the article is clear, but it really needs corrections to make the reading easier and clearer.

4. Page 3 lane 15. Does PIGF refer to placental growth factor? Please spell out all abbreviations including the figure legends.

5. The authors made an effort to develop a new system for selective denervation of the splenic nerve (SDN) by thermoablation. This effort and the alternative approach are significant and appreciated, but one wonders why not to perform a simple neurectomy of the splenic nerve as has been previously published. What is the reasoning of performing the SDN versus a conventional surgical neurectomy? What are the advantages of this new technique? Given that this technique provides the most critical results to the study, the authors should provide a deeper discussion and justification of this new technique.

6. Specify whether fig 3A refers to a bilateral or unilateral cervical vagotomy. How long before the AngII treatment was the vagotomy produced? Vagotomy induces the release of acetylcholine that will activate the splenic nerve by itself.

7. Please justify and discuss the use of CD169 in Fig 4f and 6e.
8. Supplementary Fig 5 a-c shows convincing and quantitative results that complement very well the histochemistry of Fig 6 c-e. Perhaps, Supplementary Fig 5 a-c should be included as additional panels in Fig 6.

Reviewer #4 (expert in hypertension and spleen)

Remarks to the Author:

A. Summary of the key results

Your data show that during early hypertension development, mice exhibit increased sympathetic nerve discharge to the spleen, and that this nerve discharge contributes to a subsequent systemic immune response by co-stimulating T cells. Your data also suggest that the presumptive brain-spleen pathway is realized through a vagus-to-splenic nerve connection mediated by $\alpha 7$ nAChRs. Additionally, you show that selective splenic denervation prevents hypertension development and T-cell egress from the spleen to the vasculature and kidneys.

B. Originality and interest

The basic idea is not completely new, but to my knowledge this paper represents the first direct test of the hypothesis that a brain-to-spleen pathway controlling the immune response is an important factor in the development of hypertension.

C. Data & methodology

I have a number of very significant concerns about the technical aspects of this work (though not the about the experimental design itself, which is quite good). First and foremost, I am concerned about the sympathetic nerve recordings. The records provided do not appear to be sympathetic activity. This should be pulse synchronous and of an amplitude and frequency very different from what is shown in several of your records. You should also be able to show loss of activity after ganglion blockade or during acute increases in blood pressure. Activity should markedly increase during an acute fall in blood pressure. None of this is demonstrated. And measuring baseline activity only via comparison to post-mortem values is unacceptable (since all respiratory and movement artifacts in the records will disappear post-mortem too). Finally, even if your records did show sympathetic nerve bursts, it is widely accepted in the field that it is not possible to reliably compare activity levels between animals, because of technical problems related to variability in contact of the nerve and electrode and other factors. Finally, I was unable to locate any previous papers that described and validated the technique of splenic nerve recording in mice (an extremely difficult challenge even in rats). Thus, I lack confidence in a large segment of the data presented here.

Similar uncertainties arise about your ability, for example, to selectively remove the vagal branch to the celiac ganglion. And the thermoablation method for splenic denervation needs to be described in much more detail and validation provided (beyond simply loss of splenic catecholamines). For example, how much damage is done to the splenic artery? Another worry is the very dramatic effect of splenic denervation and $\alpha 7$ nicotinic receptor knockout on blood pressure. Complete abolition of angiotensin II and DOCA-salt hypertension is striking. But this would be more convincing if full 28-day telemetric blood pressure data were provided, not simply the single before-and-after data points provided here.

D. Appropriate use of statistics and treatment of uncertainties

No problems here.

E. Conclusions

Taking the data at face value, the conclusions are reasonable and very exciting and important. However, I don't think secure conclusions are possible in light of many significant potential issues with the technical aspects of the experiments.

F.Suggested improvements
Described above.

G.References
These are fine.

H.Clarity and context
The rationale for the studies is described in a somewhat diffuse and repetitive fashion, but the main points are clear. Description of the data and the resulting conclusions are both concise and clear.

Reviewers' comments:

Reviewer #1 (expert in hypertension, sympathetic nervous system)

Remarks to the Author:

The manuscript by Carnevale D et al. show that selective splenic denervation protect against AngII-induced hypertension though the inhibition of both CD3+T cell egression and CD86 T cell expression in the spleen, and results in the inhibition of CD8+ T cell infiltration in the aorta and kidney. The results of vagotomy and $\alpha 7nAChR$ -KO suggest the complex relationships between the nervous system and immunity. This is a well-written paper containing interesting results which merit publication. However, a number of points need clarifying and certain experiments require further justification. These are given below.

We thank the Reviewer for appreciation of our work and for the points suggested below. We have performed the relative experiments that further allowed us to delineate the role of vagus-splenic connection and of $\alpha 7nAChR$ s in AngII-induced T cell egression and blood pressure rising.

1. Figure 3 show that vagotomy inhibited Ang II-induced splenic nerve activity (SSNA). Can vagotomy inhibit Ang II-induced hypertension and T cell egression?

Accordingly to Reviewer's suggestion, we have analyzed the blood pressure response to AngII and T cell egression in mice subjected to selective vagotomy. In particular, we analyzed the chronic response to AngII in mice with left celiac vagotomy, in order to test the contribution of vagal efferents on the hypertensive phenotype. As shown in **Figure 4**, we have now data reporting that celiac vagotomy hampered the typical hypertensive response to AngII and the T cell egression from spleen. Regarding the left cervical vagotomy, performed in the acute experiment shown in **Figure 3a-c**, we did not executed the long-term protocol since it has been already described that animals subjected to unilateral (left) cervical vagotomy display a reduced blood pressure 1 week after the procedure (Chen et al., *Physiol Res* 2008), because of a major contribution of vagal afferents. Thus, an eventual protection from AngII-induced hypertension, observed in mice with left cervical vagotomy, would have been misleading.

The novel data have been added in the Results section (page 6, line 24).

2. Figure 4 show that $\alpha 7nAChR$ -KO inhibited Ang II-induced SSNA and hypertension. Can $\alpha 7nAChR$ -KO inhibit Ang II-induced T cell egression?

As correctly pointed by the Reviewer, if $\alpha 7nAChR$ is needed to mediate the AngII-induced activation of splenic nerve and increase in blood pressure, the downstream signaling mediating T cell activation and egression should depend on it as well. In order to assess this point, we have performed the analysis of CD3⁺ T cells area in the white pulp of spleen from WT and $\alpha 7nAChR$ KO mice, treated with veh or AngII. As shown in **Figure 6**, we have found that lack of $\alpha 7nAChR$ blocks the T cell egression from spleen, typically induced by AngII.

The novel data have been added in the Results section (page 8, line 4).

Reviewer #2 (expert in neural control of the spleen)

Remarks to the Author:

The results show that experimental hypertension in murine models are associated with early splenic nerve discharges as measured by recording compound action potentials. These responses are modulated after vagotomy. Hypertension fails to develop in alpha-7nAChR KO animals. The authors conclude that selective splenic denervation protects against the onset of hypertension, and suggest that this is a new insight into the role and mechanism in resistant hypertension.

The findings are highly novel, timely, and should be of widespread interest to the fields of hypertension, immunology, vascular biology and neuroscience.

The approaches are valid. The data and presentation quality are extremely high. Statistical approaches are appropriate.

The data support the conclusions.

We thank this Reviewer for his/her comments that allowed us to enrich and better discuss crucial findings of the paper, like the involvement of $\alpha 7$ nAChRs in hypertension. We have accordingly modified the discussion section, as detailed below.

Minor comments:

1. The authors note that others "have demonstrated the existence of the so-called cholinergic anti-inflammatory reflex that is mediated by the celiac branch of the vagus nerve and specifically targets splenic innervation during septic challenges." It would be important to include the recent JCI paper (Vagus nerve stimulation mediates protection from kidney ischemia-reperfusion injury through $\alpha 7$ nAChR+ splenocytes <https://www.jci.org/articles/view/83658>). This a strong argument that vagus to spleen, as defined by the inflammatory reflex, and you also propose, significant and functional. It has been challenged by other groups because it is so novel, so explanation here is important.

We agree with the Reviewer about the importance of the quoted paper, recently published. This work was published just few days after our submission, and we completely agree about the relevance of the findings reported, which further argue that a vagus to spleen connection exists and is functional. Hence, we have added and discussed the reference (page 12, line 20).

2. The authors note that " AngII challenge in alpha7nAChR KO mice did not produce the increase in blood pressure typically observed in WT mice (Fig. 4g,h). Overall, these data support the existence of a vagus-splenic nerve connection that is recruited by hypertensive stimuli and mediated by alpha7nAChR." Additional explanation here is needed. Do the authors have evidence that the requirement for alpha7nAChR is in the brain? or in the ganglia? In the spleen, the acetylcholine source is from lymphocytes, so how does this affect blood pressure?

As pointed by the Reviewer, the $\alpha 7$ nAChR could be found in several tissues (brain, ganglia, immune cells, etc). Here we found that $\alpha 7$ nAChR KO mice are protected from AngII induced hypertension. The fact that $\alpha 7$ nAChR has been well characterized as the mediator of the cholinergic anti-inflammatory pathway being expressed in cytokine producing splenic macrophages (Rosas-Ballina et al., Science 2011), but also integrating the sympathetic and parasympathetic systems in presynaptic neurons of the splenic nerve (Vida et al., J Immunol 2011), pursued us to investigate where does the $\alpha 7$ nAChRs act in hypertension. Thus, main candidate hypotheses were: 1) potential role of $\alpha 7$ nAChR in immune cells (downstream to the splenic sympathetic drive); 2) potential role of $\alpha 7$ nAChR in mediating an autonomic integration among parasympathetic and sympathetic drive in the celiac ganglion (upstream to the splenic sympathetic drive). The results of marked inhibition of AngII-elicited SSNA displayed by $\alpha 7$ nAChRs KO mice supported the second hypothesis, making us propose a neuronal role of $\alpha 7$ nAChR in hypertension.

We have now detailed this piece of discussion in the revised version of the manuscript (page 12, line 11).

3. The abstract as written addresses only a small component of the major findings.... I recommend adding additional results to the abstract.

We appreciate the suggestion and we have accordingly modified the abstract, attempting to enclose all the noteworthy findings (page 2).

Reviewer #3 (expert in neural control of the spleen)

Remarks to the Author:

This is an innovative and original article showing that a cholinergic-sympathetic pathway mediated by the alpha7Receptor and the activation of splenic T cells prime immunity during hypertension. However, there are some considerations that should be clarified.

We thank the Reviewer's comments that highlighted some crucial aspects of our work. In particular, we took the opportunity to better clarify some points regarding: 1) the role of T cells in hypertension; 2) the approach of splenic denervation obtained by a novel approach of thermoablation. We have clarified these points in the specific issues.

1. The most innovative results of the article are the neuromodulation of the immune system to control hypertension through the T cells. The results correlate the AngII treatment with the splenic egression of the lymphocytes and infiltration of the CD8 T cell in the aorta. The authors are very convincing showing that those mechanisms are through the splenic nerve as demonstrated by the SDN. However, the study does not show how much the splenic egression of the T cells and the aorta infiltration contributes to Angiotensin-induced hypertension. Is this an effect contributing to hypertension too or it is just modulating immunity during hypertension? The authors should consider performing an AngII treatment in nude mice (without T cells) to determine whether T cells are required for (or contribute to) the AngII-induced hypertension.

We have focused most of our attention in this paper on the role and mechanisms of neuromodulation of the immune response, exerted by cholinergic-sympathetic nervous reflex. Our hypothesis relied on previous work aimed at dissecting the contribution of T cells in hypertension. Guzik and colleagues were the first to describe a mechanistic role of lymphocytes in hypertension (Guzik et al., JEM 2007). In particular, the authors demonstrated that mice with genetic ablation of T and B cells (RAG1 KO mice) were protected from AngII and DOCA-salt induced hypertension. More important, they found that only by restoring T cells through adoptive transfer, they were able to see again a hypertensive response, thus further demonstrating that T and not B cells are implied in raising blood pressure upon hypertensive challenges. More recently, the same group afforded the issue of characterizing the T cell subset involved in hypertension, characterizing an indispensable role of CD8 T cells for increasing blood pressure. A first set of experiments used CD8 KO mice and then they demonstrated that an adoptive transfer of activated CD8 T cells was able to increase blood pressure in mice (Trott et al., Hypertension 2014). Overall, these data support the conclusion that T cells infiltration in aorta and kidney is an indispensable mechanism for increase in blood pressure upon hypertensive challenges (reported by Harrison's lab in the publications described above and by ourselves in Carnevale et al., Immunity 2014). Thus, as suggested by the Reviewer too, T cells are necessarily required for hypertension. It is also important to notice that we have previously demonstrated that T cells egressing from spleen upon AngII challenge infiltrate aorta and kidney in a very short timing (at 3 days from AngII infusion). Indeed, we generated chimeric C57Bl/6J mice harboring spleens explanted from CD45.1 mice. This model is useful to distinguish in infiltrated tissues (i.e. aorta and kidney) CD45⁺ lineage cells coming from donor (CD45.1⁺) or recipient (CD45.2⁺) mice. By this approach, we found a significant infiltrate of T cells from donor mice in aorta and kidney after AngII, thus demonstrating that the egression of T cells is a crucial moment for allowing T cells deployment into target organs of hypertension. In the end, by demonstrating that splenectomized mice displayed a phenotype overlapping the one of RAG1 KO mice in response to AngII, being protected from hypertension, we can conclude that the immune response mediated by T cells is a process required for allowing blood pressure raising.

The background information regarding the role of T cells in hypertension has been improved in the introduction of the revised manuscript (page 3, line 7).

2. The authors have a bias referencing previous studies. Page 5, lanes 12 and 20, cite references 21 and 22 for the cholinergic- sympathetic control of the spleen. However, ref 22 does not show anything related to the spleen. Instead,

the authors appear to ignore critical references from other labs. For instance, the cholinergic control of sympathetic innervation of the spleen modulating inflammation was first reported by Huston et al (2006), J. Exp. Med. 203(7):1623-8 and confirmed later by Refs 21 and Ref 25 Vida et al. (2011) J Immunol 186, 4340-4346. Likewise, Page 10, lane 8-9 cited ref 29, for the role of T cells in the cholinergic-splenic nerve regulation of immunity, but this mechanism was first reported by Peña et al (2011) J Immunol 187(2):718-25.

We thank the Reviewer for this suggestion and we have accordingly modified the manuscript by adding the further references that allowed now to have a more comprehensive description of the existing literature on the topic.

3. In general the article is clear, but it really needs corrections to make the reading easier and clearer.

The manuscript has been reviewed by a native English speaker in the original submission and in the present form.

4. Page 3 lane 15. Does PIGF refer to placental growth factor? Please spell out all abbreviations including the figure legends.

We apologize for missing the spell out of PIGF that refers to Placental Growth Factor and we have checked for other abbreviations.

5. The authors made an effort to develop a new system for selective denervation of the splenic nerve (SDN) by thermoablation. This effort and the alternative approach are significant and appreciated, but one wonders why not to perform a simple neurectomy of the splenic nerve as has been previously published. What is the reasoning of performing the SDN versus a conventional surgical neurectomy? What are the advantages of this new technique? Given that this technique provides the most critical results to the study, the authors should provide a deeper discussion and justification of this new technique.

As correctly summarized by this Reviewer, there are previous reports in literature of simple neurectomy of the splenic nerve. The idea that we pursued was to develop a novel procedure with a high translational potential. In the field of hypertension, there is a great need of searching novel approaches for resistant patients. On this issue, in the last years great efforts have been done to develop similar approaches for renal denervation in patients (Symplivity HTN-2 Investigators. et al., Lancet 2010; Esler et al., Eur. Heart. 2014; Bhatt et al., N. Engl. J. Med. 2014). Despite initial positive and promising results, more recent clinical trials with sham control patients highlighted the necessity to explore novel mechanisms to be translated in clinical practice (Symplivity HTN-2 Investigators. et al., Lancet 2010; Esler et al., Eur. Heart. 2014; Bhatt et al., N. Engl. J. Med. 2014). Hence our idea to pursue the selective splenic denervation to target immune system in hypertension. Although in principle, a simple neurectomy could prove the same concepts and results, a selective denervation obtained with thermoablation has the potential to be translated to patients where the renal denervation strategy failed. The results obtained and presented in this paper strongly suggest that it is tempting to design dedicated approaches to resistant hypertension, based on splenic denervation even in humans. We appreciated the Reviewer's suggestion to add these considerations regarding this novel approach and we have accordingly modified the manuscript (page 10, line 14).

6. Specify whether fig 3A refers to a bilateral or unilateral cervical vagotomy. How long before the AngII treatment was the vagotomy produced? Vagotomy induces the release of acetylcholine that will activate the splenic nerve by itself.

Figure 3A refers to left cervical vagotomy, performed while recording SSNA in mice infused with AngII, starting 3 days before. We have now detailed better the experimental procedure in the methods section (page 15, line 4). In regard to the concern raised by this Reviewer about the possibility that vagotomy may induce the release of acetylcholine, thus activating the splenic nerve by itself, we have some considerations. i) We

have performed the vagotomy during the recording of SSNA in mice that received AngII in minipumps 3 days before. Accordingly, to the Reviewer's concern we should have expected an increase in SSNA. Instead, our results show that vagotomy inhibits the SSNA in mice with already activated splenic nerve. ii) In the revised version of the manuscript we have performed a further experiment to explore the chronic effect of vagotomy. Indeed, accordingly to the suggestion of Reviewer #1, we have performed the celiac vagotomy in a further group of mice, in order to analyze the blood pressure and T cell phenotypes. As shown in **Figure 4**, the celiac vagotomy protected from AngII-induced increase in blood pressure and T cell egression. This novel piece of data further support the concept that an eventual release of ACh induced by vagotomy would not affect the splenic nerve under investigation. iii) Accordingly to our findings, some data in literature reported the effect of vagotomy on ACh release, in tissues different from the splenic one, showing that the transection of cervical vagal nerves decreased dialysate ACh concentration from left ventricle myocardium (Zhan et al., *Autonomic Neurosci* 2013).

7. Please justify and discuss the use of CD169 in Fig 4f and 6e.

As better described in the revised manuscript (page 8, line 1 and 25), CD169 has been used to define the marginal zone of the spleen. Indeed, this antigen recognizes the metallophilic macrophages in the spleen, delineating a region of the organ that is exposed to body fluid. This localization is consistent with a role in antigen handling and became the focus of our attention when we found that PIGF was predominantly expressed in this zone, upon AngII challenge (Carnevale et al., *Immunity* 2014). We here show data that support the existence of a neuroimmune crosstalk realized in the marginal zone of the spleen during hypertension. In particular, **Supplementary Figure 5** and **Supplementary Figure 6**, show that TH is expressed in the same region of the spleen, hereby supporting that the focus of the neuroimmune interaction realized by hypertensive challenges is in the splenic marginal zone.

8. Supplementary Fig 5 a-c shows convincing and quantitative results that complement very well the histochemistry of Fig 6 c-e. Perhaps, Supplementary Fig 5 a-c should be included as additional panels in Fig 6.

We appreciated the Reviewer's suggestion and we have now presented all these results in regular figures (**Figures 9 and 10**).

Reviewer #4 (expert in hypertension and spleen)

Remarks to the Author:

A. Summary of the key results

Your data show that during early hypertension development, mice exhibit increased sympathetic nerve discharge to the spleen, and that this nerve discharge contributes to a subsequent systemic immune response by co-stimulating T cells. Your data also suggest that the presumptive brain-spleen pathway is realized through a vagus-to splenic nerve connection mediated by $\alpha 7nAChRs$. Additionally, you show that selective splenic denervation prevents hypertension development and T-cell egress from the spleen to the vasculature and kidneys.

B. Originality and interest

The basic idea is not completely new, but to my knowledge this paper represents the first direct test of the hypothesis that a brain-to-spleen pathway controlling the immune response is an important factor in the development of hypertension.

C. Data & methodology

I have a number of very significant concerns about the technical aspects of this work (though not the about the experimental design itself, which is quite good). First and foremost, I am concerned about the sympathetic nerve recordings. The records provided do not appear to be sympathetic activity. This should be pulse synchronous and of an amplitude and frequency very different from what is shown in several of your records. You should also be able to show loss of activity after ganglion blockade or during acute increases in blood pressure. Activity should markedly increase during an acute fall in blood pressure. None of this is demonstrated. And measuring baseline activity only via comparison to post-mortem values is unacceptable (since all respiratory and movement artifacts in the records will disappear post-mortem too). Finally, even if your records did show sympathetic nerve bursts, it is widely accepted in the field that it is not possible to reliably compare activity levels between animals, because of technical problems related to variability in contact of the nerve and electrode and other factors. Finally, I was unable to locate any previous papers that described and validated the technique of splenic nerve recording in mice (an extremely difficult challenge even in rats). Thus, I lack confidence in a large segment of the data presented here.

Similar uncertainties arise about your ability, for example, to selectively remove the vagal branch to the celiac ganglion. And the thermoablation method for splenic denervation needs to be described in much more detail and validation provided (beyond simply loss of splenic catecholamines). For example, how much damage is done to the splenic artery? Another worry is the very dramatic effect of splenic denervation and $\alpha 7$ nicotinic receptor knockout on blood pressure. Complete abolition of angiotensin II and DOCA-salt hypertension is striking. But this would be more convincing if full 28-day telemetric blood pressure data were provided, not simply the single before-and-after data points provided here.

D. Appropriate use of statistics and treatment of uncertainties

No problems here.

E. Conclusions

Taking the data at face value, the conclusions are reasonable and very exciting and important. However, I don't think secure conclusions are possible in light of many significant potential issues with the technical aspects of the experiments.

F. Suggested improvements

Described above.

G. References

These are fine.

H. Clarity and context

The rationale for the studies is described in a somewhat diffuse and repetitive fashion, but the main points are clear. Description of the data and the resulting conclusions are both concise and clear.

We thank the comments and criticisms raised by this Reviewer that gave us the possibility to clarify further technical aspects that were overlooked in the first version of the manuscript.

The major concern of this Reviewer seems related to the possibility to approach the splenic district in the mouse. We are aware that this is a great experimental challenge and that few papers in literature present results obtained in this setting. However, we would like to pose the attention of the Reviewer to the fact that we have already published a challenging approach to the splenic district, i.e. the realization of spleen transplantation in mice (Carnevale et al., *Immunity* 2014). It is also noteworthy to remark that the same approach was previously published in important papers from another group (Swirski et al., *Science* 2009), thus further supporting that approaching this district is not impossible, although extremely difficult. Here following we discuss the single technical points raised: 1) sympathetic nerve recording; 2) splenic denervation.

1) In relation to the concern raised by the Reviewer that we are really recording the splenic sympathetic nerve activity, we would like to discuss better the technical issues related to this procedure and to its establishment. We agree with the Reviewer's considerations about the characteristics that a sympathetic nerve recording should display (1. pulse synchronous, 2. reduction during increases in blood pressure, 3. increase during fall in blood pressure, 4. loss of activity during ganglion blockade). In the previous version of the manuscript, we gave just the results concerning the experimental setting of interest, while omitting the data related to the establishment of the procedure itself and thus the points summarized by the Reviewer. We are now providing the whole data in **Supplementary Figure 1** and **Supplementary Figure 2**, showing that:

a) Splenic sympathetic nerve recording is pulse synchronous. In particular, the majority of nerve burst can be identified as a cluster of action potential that are synchronized to the raw blood pressure signal. Indeed, representative recordings of SSNA show bursts after the peak in systolic blood pressure. As previously reported in reliable pieces of literature (Stocker and Muntzel, *Am J Physiol Heart Circ Physiol* 2013; Guild et al., *Exp Physiol* 2010; Malpas, *Physiol Rev* 2010), the recordings from each nerve district has a characteristic pattern of activity, being the renal nerve entirely rhythmic and the lumbar and splanchnic semirhythmic. In our paper, we provide results showing that the splenic nerve activity has a behavior comparable to the latter nerves.

b) In order to verify that we were measuring true sympathetic nerve activity, we manipulated blood pressure with sodium nitroprusside (**Supplementary Figure 1**) and phenylephrine (**Supplementary Figure 2**). As expected, SSNA was markedly stimulated in response to the reduction in blood pressure obtained by sodium nitroprusside and was inhibited after the increase in blood pressure determined by phenylephrine bolus.

c) Ganglionic blockade with hexamethonium completely inhibited sympathetic bursts of the splenic nerve (**Supplementary Figure 1** and in **Supplementary Figure 2**). As shown by the representative recordings obtained with ganglionic blockade and during post mortem, no significant respiratory and movement artifacts were observable, thus supporting the choice to measure the noise of each recording with the post mortem. As discussed by the Reviewer, the baseline activity is often evaluated after ganglionic blockade of mice. However, in our specific case where a great part of the study has been focused on nACh receptors and, in particular, on $\alpha 7$ nAChRs, we preferred to avoid the use of hexamethonium, a pharmacological agent directly impacting on the nAChRs, the molecular mechanism under investigation. On the other hand, the use of post mortem recording has been considered as well as a methodological approach to correct the signal to background noise (Hamza and Hall, *Hypertension* 2012).

d) The Reviewer is concerned about our choice to compare activity levels between animals with absolute data, instead of % of variation. We are aware that this latter is the most common procedure used in the analysis. However, there are at least two issues that should be noticed. 1) In the vast majority of the experimental settings reported in literature, the focus of analysis is referred to the variation of a response respect to a baseline activity in the same animal. Instead, we mostly focused our experiments on the comparison among different groups of mice (different for genotype, treatment etc.). In this case, it would have been more complicated to express the % SSNA in a group of mice as compared to another control group. 2) The choice to express responses as % change vs baseline is commonly used to correct for technical problems related to variability in the animal preparation (as stated by the Reviewer). We have reported

absolute data of each result, showing how the variability among mice is low. Accordingly to these considerations, we believe that our choice to report data as absolute values of SSNA instead of a % of variation, further strengthen our results.

2) The Reviewer has raised the same kind of concern when discussing the results obtained with the removal of vagal branch of celiac ganglion and selective denervation of splenic nerve. As previously stated, we have published results regarding spleen transplantation in mice, an even more challenging experimental setting. We believe that, with that experience, we could have faced the technical issue to approach the splenic nerve, having the required microsurgery expertise. However, the concern raised by the Reviewer was a great opportunity for us to refine further the methodology developed by establishing an additional technique to evaluate the positive outcome of the procedure in terms of vitality and function of the denervated artery and spleen. In particular, we are showing data reporting unaltered pulse wave of splenic artery, as evaluated by ultrasound imaging (**Figure 7**), and capability to correctly perfuse the spleen, as shown by microCT angiography and tissue analysis (**Figure 7**).

In the revised figures, we are also giving the full time-course analysis of blood pressure recordings.

In the end, although there are some papers that afford the recording of splenic nerve activity in rats (Ganta, C.K., et al. *Am. J. Physiol. Heart Circ. Physiol.* 2005; Ganta, C.K., et al. *Am J Physiol Regul Integr Comp Physiol.* 2006; Tanida, M., et al. *Neurosci. Lett.* 2016) and very few in mice (Izumo, T., et al. *NeuroReport* 2013; Horii, Y. et al. *Neurosci. Lett.* 2012), we agree with the Reviewer that it is difficult (probably impossible) to locate in the literature papers aimed at investigating the issue of splenic nerve activity in the context of cardiovascular diseases. We believe that the data presented here, although deserved some skepticism in the Reviewer, should be considered a pioneering approach that will allow future further investigation in the emerging field of researches exploring immune mechanisms in cardiovascular diseases.

REVIEWERS' COMMENTS:

Reviewer #1 (Remarks to the Author):

The authors well responded to my comments. Results of the additional experiments strengthen their hypothesis. This revised manuscript is now acceptable for Nature Communications.

Reviewer #2 (Remarks to the Author):

The authors show that hypertensive challenges activate sympathetic nerve discharge in the spleen via a vagus-splenic nerve circuit mediated by nicotinic cholinergic receptors. The sympathetic discharge induced by hypertensive stimuli was absent in celiac vagotomized mice, as well as in $\alpha 7$ nAChR KO mice. This cholinergic-sympathetic pathway permits T cell costimulation and aggression upon hypertensive challenges.

They propose a novel experimental procedure for selective splenic denervation, which protects against hypertension.

The findings are highly original, and will be of great interest to the fields of neuroscience, immunology and hypertension.

The results represent the product of a technical tour de force, and set new standards for inquiries of this kind. The statistics are appropriate, and the data support the conclusions.

Major comments:

1. The authors have replied in detail to my prior questions, and moreover, have provided additional data supporting the original paper and improving it greatly in the process.

2. One final comment is a recommendation to use more precise terminology regarding "sympathetic" nerves. The splenic nerve is a mixed nerve, and "sympathetic" lacks precision. Better to say "splenic nerve" as the anatomic usage; and "adrenergic" nerve when referring to the function. This is actually very very important to the reader, because the concepts presented here give further new evidence that the division between "sympathetic" and "parasympathetic" is not absolute as taught in old textbooks. Rather, as Henry Dale caution 100 years ago, we should describe nerves by their neurotransmitter and anatomy, not by a broad functional classification. Hence "splenic" or "adrenergic."

Reviewer #3 (Remarks to the Author):

This is an innovative, original and sound article showing that the vagal-celiac-splenic neuronal network mediated by the $\alpha 7$ Receptor is critical for the activation of splenic T cells egression during hypertension. The scientific methodology and reasoning are sound and the overwhelming number of results are convincing and supporting the conclusions.

I thank the authors for their effort to address my concerns and improve the article. However, even the results are clear, several sections of the manuscript are confusing and difficult to understand. I have to insist in a review of the writing of the manuscript. I understand that the "The manuscript has been reviewed by a native English speaker". I wonder whether this reviewer has scientific background. Expressions like "older papers", or sentences in the abstract like "Additionally, we propose a novel

experimental procedure of selective splenic denervation, which protects against hypertension, thereby unveiling a great translational potential of our findings, shedding new perspectives on the use of sympathetic denervation in resistant hypertension" do not help the profile of readers of the journal. I am thinking in the broad spectrum of readers of the journal that may not have a significant background in neuro-modulation. A more direct and clear style will not only increase the significance of the article but also the general impact of this study on the typical reader of the journal. Below, I propose a few examples of suggestions for the perusal of the authors.
Abstract page 2 line 3-4 Review "how this connection is realized remains unknown" for "The physiological and molecular mechanisms of this connection are unknown".

Abstract page 2 line 4 "early activate" for "activate". Remove "early"

Abstract page 2 line 6 Remove "In fact"

Abstract page 2 line 8 change "Moreover, we reveal that this cholinergic-sympathetic pathway permits T cell costimulation...." for "This cholinergic-sympathetic pathway is necessary for T cell activation"

Abstract page 2 line 10 Split sentence : "Additionally, we propose a novel experimental procedure of selective splenic denervation, which protects against hypertension, thereby unveiling a great translational potential of our findings, shedding new perspectives on the use of sympathetic denervation in resistant hypertension" for "Additionally, we also show that selective thermoablation of the splenic nerve prevents T cell egression and protects against hypertension. This novel experimental procedure for selective splenic denervation support new clinical strategies for resistant hypertension."

Page 3 line 20 change "older papers..." for "preliminary studies..."

Page 4 line 5 change "We do know the chronic effects of AngII are necessarily 5 mediated by actions coming from the brain" for "The chronic effects of AngII are mediated by signals from the brain mediated the by peripheral networks"

Page 5 line 25 change "A vagus-splenic nerve connection realizes a cholinergic-sympathetic drive in hypertension" for "The vagal-celiac-splenic nerve connection mediates the induction of hypertension"

Page 6 line 11 change "how intact vagus nerve signaling could affect splenic ..." for "how vagal signals could affect splenic....."

Page 6 line 12 change "In the absence of any afferent or efferent effects of the vagus nerve, no SSNA was anymore detectable in mice infused with AngII (Fig. 3b)" for "Cervical vagotomy completely prevented the potential AngII to activate the splenic sympathetic nerve (Fig. 3b). These results indicate that the AngII-induced activation of the splenic nerve is mediated by the cervical vagus nerve"

Page 7 line 7 Change "7nAChR mediates the vagus-splenic drive activated by hypertensive challenge and is crucial for 7 blood pressure regulation" for "7nAChR mediates the vagal-celiac-splenic nerve connection inducing hypertension"

Page 8 line 9 Change "Selective splenic denervation protects against onset of hypertension and prevents T cell egression and 9 deployment toward target organs" for "Selective thermoablation of the splenic nerve prevents T cell egression and protects against hypertension"

Page 10 line 8 Change "Consistent with findings obtained during endotoxemia, we further found this

interaction to be mediated at the molecular level by $\alpha 7$ nAChR" for "Our results also indicate that the vagal-celiac-splenic nerve connection regulating blood pressure is mediated by the $\alpha 7$ nAChR, similar as previously described in endotoxemia (26). This vagal-celiac-splenic network regulates T cell activation in endotoxemia and septic shock (33,34). Our results now indicate that a similar mechanism induces T cell egression contributing to hypertension."

Page 12 line 16. This is a wrong citation. "also integrating the sympathetic and parasympathetic 16 systems in presynaptic neurons of the splenic nerve (25)...." Reference 25 does not analyze $\alpha 7$ Receptors. This will probably refer to reference 26.

REVIEWERS' COMMENTS:

Reviewer #1 (Remarks to the Author):

The authors well responded to my comments. Results of the additional experiments strengthen their hypothesis. This revised manuscript is now acceptable for Nature Communications.

Thank you.

Reviewer #2 (Remarks to the Author):

The authors show that hypertensive challenges activate sympathetic nerve discharge in the spleen via a vagus-splenic nerve circuit mediated by nicotinic cholinergic receptors. The sympathetic discharge induced by hypertensive stimuli was absent in celiac vagotomized mice, as well as in $\alpha 7$ nAChR KO mi. This cholinergic-sympathetic pathway permits T cell costimulation and aggression upon hypertensive challenges.

They propose a novel experimental procedure for selective splenic denervation, which protects against hypertension.

The findings are highly original, and will be of great interest to the fields of neuroscience, immunology and hypertension.

The results represent the product of a technical tour de force, and set new standards for inquiries of this kind. The statistics are appropriate, and the data support the conclusions.

Major comments:

1. The authors have replied in detail to my prior questions, and moreover, have provided additional data supporting the original paper and improving it greatly in the process.

2. One final comment is a recommendation to use more precise terminology regarding "sympathetic" nerves. The splenic nerve is a mixed nerve, and "sympathetic" lacks precision. Better to say "splenic nerve" as the anatomic usage; and "adrenergic" nerve when referring to the function. This is actually very very important to the reader, because the concepts presented here give further new evidence that the division between "sympathetic" and "parasympathetic" is not absolute as taught in old textbooks. Rather, as Henry Dale caution 100 years ago, we should describe nerves by their neurotransmitter and anatomy, not by a broad functional classification. Hence "splenic" or "adrenergic."

We thank this Reviewer for the positive comments and the further suggestion to use a more precise definition of the splenic nerve when referring to the anatomic context (splenic nerve) or the related function (adrenergic). The manuscript has been accordingly modified when the definition could be confounding for the reader.

Reviewer #3 (Remarks to the Author):

This is an innovative, original and sound article showing that the vagal-celiac-splenic neuronal network mediated by the $\alpha 7$ Receptor is critical for the activation of splenic T cells egression during hypertension. The scientific methodology and reasoning are sound and the overwhelming number of results are convincing and supporting the conclusions.

I thank the authors for their effort to address my concerns and improve the article. However, even the results are clear, several sections of the manuscript are confusing and difficult to understand. I have to insist in a review of the writing of the manuscript. I understand that the "The manuscript has been reviewed by a native English speaker". I wonder whether this reviewer has scientific background. Expressions like "older papers", or sentences in the abstract like "Additionally, we propose a novel experimental procedure of selective splenic denervation, which protects against hypertension, thereby unveiling a great translational potential of our findings, shedding new perspectives on the use of sympathetic denervation in resistant hypertension" do not help the profile of readers of the journal.

I am thinking in the broad spectrum of readers of the journal that may not have a significant background in neuro-modulation. A more direct and clear style will not only increase the significance of the article but also the general impact of this study on the typical reader of the journal. Below, I propose a few examples of suggestions for the perusal of the authors.

We thank the Reviewer for his/her helpful suggestions and we have accepted all of them when possible. The only exclusions were due to the character restriction guidelines for subheadings that did not allow us to accept all proposals. However, we asked for a final editing of our manuscript that was revised by a native English speaker.

Abstract page 2 line 3-4 Review "how this connection is realized remains unknown" for "The physiological and molecular mechanisms of this connection are unknown".

Abstract page 2 line 4 "early activate" for "activate". Remove "early"

Abstract page 2 line 6 Remove "In fact"

Abstract page 2 line 8 change "Moreover, we reveal that this cholinergic-sympathetic pathway permits T cell costimulation...." for "This cholinergic-sympathetic pathway is necessary for T cell activation"

Abstract page 2 line 10 Split sentence : "Additionally, we propose a novel experimental procedure of selective splenic denervation, which protects against hypertension, thereby unveiling a great translational potential of our findings, shedding new perspectives on the use of sympathetic denervation in resistant hypertension" for "Additionally, we also show that selective thermoablation of the splenic nerve prevents T cell egression and protects against hypertension. This novel experimental procedure for selective splenic denervation support new clinical strategies for resistant hypertension."

Page 3 line 20 change "older papers..." for "preliminary studies..."

Page 4 line 5 change "We do know the chronic effects of AngII are necessarily 5 mediated by actions coming from the brain" for "The chronic effects of AngII are mediated by signals from the brain mediated the by peripheral networks"

Page 5 line 25 change "A vagus-splenic nerve connection realizes a cholinergic-sympathetic drive in hypertension" for "The vagal-celiac-splenic nerve connection mediates the induction of hypertension"

Page 6 line 11 change "how intact vagus nerve signaling could affect splenic ..." for "how vagal signals could affect splenic....."

Page 6 line 12 change "In the absence of any afferent or efferent effects of the vagus nerve, no SSNA was anymore detectable in mice infused with AngII (Fig. 3b)" for "Cervical vagotomy completely prevented the potential AngII to activate the splenic sympathetic nerve (Fig. 3b). These results indicate that the AngII-induced activation of the splenic nerve is mediated by the cervical vagus nerve"

Page 7 line 7 Change "7nAChR mediates the vagus-splenic drive activated by hypertensive challenge and is crucial for 7 blood pressure regulation" for "7nAChR mediates the vagal-celiac-splenic nerve connection inducing hypertension"

Page 8 line 9 Change "Selective splenic denervation protects against onset of hypertension and prevents T cell egression and 9 deployment toward target organs" for "Selective thermoablation of the splenic nerve prevents T cell egression and protects against hypertension"

Page 10 line 8 Change "Consistent with findings obtained during endotoxemia, we further found this interaction to be mediated at the molecular level by $\alpha 7$ nAChR" for "Our results also indicate that the vagal-celiac-splenic nerve connection regulating blood pressure is mediated by the $\alpha 7$ nAChR, similar as previously described in endotoxemia (26). This vagal-celiac-splenic network regulates T cell activation in endotoxemia and septic shock (33,34). Our results now indicate that a similar mechanism induces T cell egression contributing to hypertension."

Page 12 line 16. This is a wrong citation. "also integrating the sympathetic and parasympathetic 16 systems in presynaptic neurons of the splenic nerve (25)...." Reference 25 does not analyze $\alpha 7$ Receptors. This will probably refer to reference 26. Reviewer #3 (expert in neural control of the spleen)